# New techniques for gap-filling and partitioning of $H_2O$ and $CO_2$ eddy fluxes measured over forests in complex mountainous terrain

Minseok Kang[1], Joon Kim[1,2,3,4], Bindu Malla Thakuri[2], Junghwa Chun[5], Chunho Cho[6]

[1]National Center for AgroMeteorology, Seoul, 08826, South Korea
[2]Program in Rural Systems Engineering, Department of Landscape Architecture & Rural Systems Engineering, Seoul National University, Seoul, 08826, South Korea
[3]Interdisciplinary Program in Agricultural & Forest Meteorology, Seoul National University, Seoul, 08826, South Korea
[4]Institute of Green Bio Science and Technology, Seoul National University Pyeongchang Campus, Pyeongchang, 25354, South Korea
[5]Department of Forest Conservation, National Institute of Forest Science, Seoul, 02455, South Korea
[6]National Institute of Meteorological Research, Seogwipo, 63568, South Korea

*Correspondence to*: Minseok Kang (ms-kang@ncam.kr)

**Abstract.** The continuous measurement of $H_2O$ and $CO_2$ fluxes using the eddy covariance (EC) technique is still challenging for forests in complex terrain because of large amounts of wet canopy evaporation ($E_{WC}$), which occur during and following rain events when the EC systems rarely work correctly, and the horizontal advection of $CO_2$ generated at night. We propose new techniques for gap-filling and partitioning of the $H_2O$ and $CO_2$ fluxes: (1) a model-stats hybrid method (MSH) and (2) a modified moving point test method ($MPT_m$). The former enables the recovery of the missing $E_{WC}$ in the traditional gap-filling method and the partitioning of the evapotranspiration (ET) into transpiration and (wet canopy) evaporation. The latter determines the friction velocity ($u^*$) threshold based on an iterative approach using moving windows for both time and $u^*$, thereby allowing not only the nighttime $CO_2$ flux correction and partitioning but also the assessment of the significance of the $CO_2$ drainage. We tested and validated these new methods using the datasets from two flux towers, which are located at forests in hilly and complex terrains. The MSH reasonably recovered the missing $E_{WC}$ of 16 ~ 41 mm year$^{-1}$ and separated it from the ET (14 ~ 23% of the annual ET). The $MPT_m$ produced consistent carbon budgets using those from the previous research and diameter increment, while it has improved applicability. Additionally, we illustrated certain advantages of the proposed techniques, which enables us to understand better how ET responses to environmental changes and how the water cycle is connected to the carbon cycle in a forest ecosystem.

## 1 Introduction

Forest ecosystems share three properties that are significant in their interactions with the atmosphere. They are extensive, dense and tall, and thus produce sizable aerodynamic roughness and canopy storage for rainfall interception/evaporation (e.g., Shuttleworth, 1989). Since most of the flat terrains are used as agricultural lands and towns (or cities), substantial areas of

forest exist in mountainous terrains where the fundamental assumptions of eddy covariance (EC) measurement (flat and homogeneous site, e.g., Baldocchi et al., 1988) are violated. These facts hinder the use of the EC method from assessing the net ecosystem exchanges (NEE) of $H_2O$ and $CO_2$ in forests.

Considering that EC measures compound 'net' fluxes and its gaps are unavoidable, we commonly take great care for flux gap-filling and partitioning. Basically, the gap-filling and partitioning are a kind of interpolation and extrapolation based on that EC measurement has high temporal resolution and the bio-meteorological processes is a (repetitive) cycle ("redundancy" of data) (Papale et al., 2012). Generally, they consist of the following procedure: (1) setting a target flux (e.g., $CO_2$/$H_2O$/$CH_4$ fluxes, ecosystem respiration), (2) selecting drivers which control the target flux, (3) identifying relationships between the (appropriate) target flux (which can represent true NEE) and the drivers, and (4) interpolating and extrapolating the relationships during a certain period when the relationship is maintained (e.g., Papale et al., 2012; Reichstein et al., 2012). In this context, the gap-filling and partitioning (including nighttime $CO_2$ flux correction) are coterminous with each other. The related scientific issue is determining/selecting the number and type of drivers, and the method and the time window size to identify the relationship. It depends on data availability, temporal scale of the process, and ecosystem state change. Those processes require extra care for the measurement in complex mountainous terrain as follows.

**1.1 Wet canopy evaporation: gap-filling and partitioning**

Wet canopy evaporation ($E_{WC}$) is an evaporation of the intercepted water by the vegetation canopy during and following rain events, which may consist of a significant portion of evapotranspiration (ET). Over forests, it is hard to measure the $E_{WC}$ primarily due to the malfunction of an open-path EC system with rainfall. Although a closed-path system with an intake tube enables the $E_{WC}$ measurement in the rain, the attenuation of the turbulent flow inside the tube acts as a low-pass filtering, which results in a significant underestimation of the $E_{WC}$. Furthermore, the attenuation domain expands with an increasing relative humidity (RH) from high frequency to medium frequency (e.g., Ibrom et al., 2007; Fratini et al., 2012). The closed-path EC system with the heated tube may be the most appropriate for measuring ET in the rain (e.g., Goodrich et al., 2016). The missing (or low quality) data can be gap-filled using general gap-filling methods such as the marginal distribution sampling and artificial neural network (e.g., Reichstein et al., 2005; Papale and Valentini 2003). However, Kang et al. (2012) showed that, without a proper consideration of the $E_{WC}$, such gap-filled ET data under the wet canopy conditions are underestimated because the data used in such gap-filling are mostly collected dry or partially wet canopy condition when the EC systems work properly. The authors proposed an improved gap-filing method that is coupled with a simple canopy (water) interception model.

The ET represents a combination of the $E_{WC}$, transpiration ($T$), and soil evaporation ($E_S$), which are controlled by different mechanisms and processes. Therefore, the partitioning of ET into the $E_{WC}$, $T$, and $E_S$ is required to understand how ET is regulated by environmental changes and the how water cycle is connected to the carbon cycle in a forest ecosystem. For these reasons, there have been many previous studies that partition ET using other supplementary measurements or empirical/process models (e.g., Wilson et al., 2001; Yepez et al., 2003; Daikoku et al., 2008; Stoy et al., 2006; Hu et al.,

2009; Kang et al., 2009b). Despite the many previous studies on ET partitioning, most of them have focused on the partitioning of ET into the $E_S$ (or direct evaporation, i.e., a sum of $E_S$ and $E_{WC}$) and $T$. In the case of forest ecosystems with a dense canopy under a monsoon climate (e.g., East Asia, South Asia), the $E_{WC}$ can play a greater role than the $E_S$. In this context, it is necessary to pay attention to the method described by Kang et al. (2012), which not only allows the proper estimation and gap-filling of the missing evaporation data under wet canopy conditions but also enables the partitioning of ET into the $E_{WC}$ and $T$ appropriately after certain modifications.

## 1.2 Nighttime CO$_2$ flux: correction for complex topography

In EC measurements over complex mountainous terrain, the nighttime CO$_2$ flux correction (i.e., eliminate the advection-affected data and fill the gaps) is one of the most important and challenging tasks. There are two types of widely used methods: (1) the friction velocity ($u^*$) filtering method, and (2) the advection-based filtering method. The most commonly used method is the $u^*$ filtering method that optimizes the parameters of the ecosystem respiration (RE) function using the observed nighttime CO$_2$ flux when $u^*$ is greater than a threshold, (i.e., there is no dependency of the nighttime CO$_2$ flux on $u^*$) (e.g., Falge et al., 2001; Gu et al., 2005). The filtered data are replaced with the estimated data using ecosystem temperatures and the RE function with the optimized parameters (which is optimized using the remaining data). The $u^*$ filtering method cannot be applied at sites where the $u^*$ threshold cannot be determined and/or the drainage flow is generated during most of the night. Accordingly, an advection-based method was developed for hilly terrain sites that are affected by drainage flow (van Gorsel et al., 2007, 2008 and 2009). It omits most of the nighttime data and uses the observed CO$_2$ flux data near sunset when the nighttime advection has not yet affected to EC measurement. The nighttime correction is also important for the partitioning of NEE into gross primary productivity (GPP) and RE because the nighttime RE-temperature relationship is used to extrapolate the daytime RE (e.g., Reichstein et al., 2005).

The Gwangneung deciduous and coniferous forests sites in Korea, which are a part of Korea Flux Monitoring Network (KoFlux), are typical sites situated in hilly and complex terrains where these methods are difficult to apply appropriately because the CO$_2$ drainage generated earlier than the time assumed by the method. Kang et al. (2017) developed the site-specific quality control filter to exclude the data strongly affected by CO$_2$ advection. The filter identifies the observations that occur when a strong information flow toward the bottom of the slope exists in the dynamical process network of the observed multi-level CO$_2$ concentrations. This site-specific filter, which is qualitatively similar to the application of the traditional correction methods in a hybrid way, substantially reduced the disagreement among the three different conventional methods of nighttime CO$_2$ corrections. However, this method has low applicability because the measurements of CO$_2$ concentration profiles are required at two or more locations along the drainage, and a long time series data is necessary to produce robust results. To overcome these shortcomings, we propose a hybrid of the $u^*$ filtering and advection-based methods, which can be applied unconditionally, "*everywhere, all of the time.*"

### 1.3 Overview of research

In this study, we propose new techniques for gap-filling and partitioning of the $H_2O$ and $CO_2$ fluxes measured over forests in complex mountainous terrain. First, we introduce a model-stats hybrid method, which can not only recover the missing $E_{WC}$ in the general gap-filling method but also separate it from ET. Then, an automated statistical method is introduced to determine the $u^*$ threshold based on an iterative approach using a moving window of both time and $u^*$ (i.e., the modified moving point test method). This method enables the determination of 'the significance of $CO_2$ drainage and the $u^*$ threshold for the nighttime $CO_2$ flux correction and partitioning. We tested and validated these new methods using the datasets from the two flux towers, which are located in forests with hilly and complex terrains. Additionally, we illustrated certain advantages of the new techniques.

### 2 Materials and Methods

#### 2.1 Study sites

In the Gwangneung National Arboretum, there are two eddy covariance flux towers: the Gwangneung deciduous forest located at the top of a hill (GDK; 37° 45' 25" N, 127° 09' 12" E) and the Gwangneung coniferous forest located at the bottom (GCK; 37° 44' 54" N, 127° 09' 45" E). Gwangneung has been protected to minimize human disturbance over the last 500 years. Both sites are located on complex, hilly catchment with a mean slope of 10 – 20°. The two towers are ~ 1.2 km apart, and the mean slope between them is ~ 6.2° (Moon et al., 2005). The east/west slopes are gentle, whereas the north/south slopes are steep in the catchment. The mountain-valley circulation is dominant wind regime in the sites (Hong et al., 2005; Yuan et al., 2007). Meteorological records from an automatic weather station ~ 1.6 km northeast of the tower for 1997-2016 show that annual mean air temperature is 10.1±0.6°C and the mean precipitation is 1,472±352 mm (National Climate Data Service System, http://sts.kma.go.kr/). At the GDK site, the vegetation is dominated by an old natural forest of *Quercus* sp. and *Carpinus* sp. (80 – 200 years old) with a mean canopy height of ~ 18 m and a maximum leaf area index (LAI) of ~ 6 $m^2$ $m^{-2}$ in June. Compared to the GDK site, the GCK site is in a lower area and is a flat, plantation forest with the dominant species of *Abies holophylla* (approximately 80 years old) with a mean canopy height of ~ 23 m and a maximum LAI of ~ 8 $m^2$ $m^{-2}$ in June. Further descriptions of the sites can be found in Kim et al. (2006) and Kang et al. (2017).

#### 2.2 Measurements and data processing

The $H_2O$ and $CO_2$ fluxes have been measured since 2006 and 2007 at the GDK site and GCK site, respectively. At both sites, the EC system was used to measure the fluxes from a 40 m tower. The wind speed and temperature were measured with a three-dimensional sonic anemometer (Model CSAT3, Campbell Scientific Inc., Logan, Utah, USA), while the $H_2O$ and $CO_2$ concentrations were measured with an open-path infrared gas analyzer (IRGA; Model LI-7500, LI-COR Inc., Lincoln, Nebraska, USA) at both sites. Half-hourly ECs and the associated statistics were calculated online from the 10 Hz raw data

and stored in dataloggers (Model CR5000, Campbell Scientific Inc.). Other measurements such as net radiation, air temperature, humidity, and precipitation were sampled every second, averaged over 30 minutes, and logged in the dataloggers (Model CR3000 for the GDK site and CR1000 for the GCK site, Campbell Scientific Inc.). More information regarding the EC and meteorological measurements can be found in Kwon et al. (2009), and Kang et al. (2009a).

The multi-level profile systems were installed to measure the vertical profiles of the $CO_2$ and $H_2O$ concentrations at both sites and to estimate the storage flux using a closed-path IRGA (Model: LI-6262, LI-COR Inc.). The measurement heights were 0.1, 1, 4, 8 (base of the crown), 12 (middle of the crown), 18 (the canopy top), 30, and 40 m for the GDK site and 0.1, 1, 4, 12 (base of the crown), 20 (middle of the crown), 23 (the canopy top), 30, and 40 m for the GCK site. More information regarding the multi-level profile system can be found in Hong et al. (2008) and Yoo et al. (2009).

To improve the data quality, the collected data were examined by the quality control procedure based on the KoFlux data processing protocol (Hong et al., 2009; Kang et al., 2014). This procedure includes a sector-wise planar fit rotation (PFR; Wilczak et al., 2001; Yuan et al., 2007; Yuan et al., 2011), the WPL (Webb-Pearman-Leuning) correction (Webb et al., 1980), a storage term calculation (Papale et al., 2006; see Appendix A for more details regarding the storage term calculation), spike detection (Papale et al., 2006), gap-filling (marginal distribution sampling method; Reichstein et al., 2005), and nighttime $CO_2$ flux correction (van Gorsel et al., 2009). The details of the gap-filling and partitioning methods are described in the next chapters.

## 2.3 Gap-filling and partitioning methods for the $H_2O$ flux

### 2.3.1 Marginal distribution sampling (MDS) method

The missing $H_2O$ flux (i.e., evapotranspiration, ET) data were gap-filled using the marginal distribution sampling (MDS)
method (Reichstein et al., 2005; Hong et al., 2009). This method calculates a median of ET under similar meteorological conditions within a time window of 14 days and replaces the missing values with the median. The intervals of the similar meteorological conditions were 50 W m$^{-2}$ for the downward shortwave radiation ($R_{sdn}$), 2.5°C for the air temperature ($T_a$), and 5.0 hPa for the vapor pressure deficit (VPD). If similar meteorological conditions were unavailable within the time window, its interval increased in increments of 7 days before and after the missing data point (i.e., 14 days of window size)
until it reached 56 days (i.e., before and after 7 days → 14 days → 21 days → 28 days). When the missing ET values could not be filled in a time window less than 56 days, $R_{sdn}$ was exclusively used following the same approach (i.e., calculating a median of ET under similar $R_{sdn}$ conditions within a time window). This gap-filling method is used for not only the $H_2O$ flux but also the sensible heat and daytime $CO_2$ fluxes.

### 2.3.2 Modeling of wet canopy evaporation

For estimating the wet canopy evaporation ($E_{WC}$), a simplified version of the Rutter sparse model (Valente et al., 1997) included in the VIC LSM (Variable Infiltration Capacity Land Surface Model, Liang et al., 1994) was used in the KoFlux data processing program. The $E_{WC}$ is estimated as follows:

$$E_{WC\_Mod} = \sigma_f E_p \left( \frac{W_c}{S} \right)^n \left( \frac{r_a}{r_a + r_0} \right)$$

(1)

where $E_{WC\_Mod}$ is the modeled $E_{WC}$, $\sigma_f$ is the vegetation fraction (i.e., 1–gap fraction); and $E_p$ is the potential evaporation

$$E_p = \frac{\varepsilon A + \rho c_p \cdot VPD \cdot g_a / \gamma}{\lambda(\varepsilon + 1)}$$

(                                                                                      where $\varepsilon$ is the dimensionless ratio of the slope of the saturation vapor pressure curve to the psychrometric constant $\gamma$, $A$ is the available energy, $\rho$ is the density of air, $c_p$ is the specific heat of air, $g_a$ is the aerodynamic conductance ($= 1/ r_a$), $\lambda$ is the latent heat of vaporization); $r_a$ is the aerodynamic resistance to heat and water

vapor transport; $S$ is the canopy storage capacity; and $r_0$ is the architectural resistance. The term, ($r_a / (r_a + r_0)$), is added to consider the variation of the gradient of specific humidity between the leaves and the overlying air in the canopy layer. $W_c$ is the intercepted canopy water, and the exponent $n$ is an empirical coefficient.

$W_c$ is estimated as:

$$\frac{\partial W_c}{\partial t} = \sigma_f P - D - E_{WC\_Mod}$$

(2)

where $P$ is the input total rainfall and $D$ is the drip. When $W_c > 0$, the canopies are wet. When $W_c > S$, the drip starts ($D > 0$). There are many inputs (i.e., $E_p$ and $P$) and parameters (i.e., $\sigma_f$, $S$, $n$, $r_a$, and $r_0$) for estimating the $E_{WC}$ and $W_c$. $E_p$, $P$, and $r_a$

($= r_{am} + r_b$ where $r_{am}$ and $r_b$ are the aerodynamic resistance of momentum transfer and the excess resistance; $r_{am} = \overline{U} / u_*^2$

where $\overline{U}$ is the wind speed, $u^*$ is the friction velocity; $r_b \approx \dfrac{4.63}{u_*}$ ; Thom, 1972; Kim and Verma, 1990; Kang et al., 2009a)

can be obtained/estimated from the flux tower measurement. The parameters can be divided into constant parameters (i.e., $n$

and $r_0$) and seasonally varied parameters (i.e., $\sigma_f$ and $S$). The default values (before optimization) of $n$ and $r_0$ are 2/3 and 2 s m$^{-1}$, respectively. $\sigma_f$ ($= 1 - $ gap fraction) and $S$ are functions of LAI (leaf area index): (1) the gap fraction is estimated by exp($-k \times$ LAI), where $k$ varies from 0.3 to 1.5, depending on the species and canopy structure (Jones, 2013, $k = 0.75$ and 0.485 for the GDK and GCK, respectively; (2) $S$ is estimated by $K_L \times$ LAI, where $K_L$ varies from 0.1 to 0.3 (default value of $K_L = 0.2$, see Appendix B for more details). $\sigma_f$ and LAI can be obtained from a plant canopy analyzer or digital photography

(e.g., Macfarlane et al., 2007, Hwang et al., 2016). If actual measurement is not available, MODIS (moderate-resolution imaging spectroradiometer) LAI can be used alternatively. In this study, $\sigma_f$ (actually $k$) and LAI were estimated using a plant canopy analyzer (Model LAI-2000; Li-Cor Inc.).

The generalization of the model can be augmented by providing the parameter optimization procedure using available flux data under wet canopy condition. We argue that this is better than the validation using other datasets because the parameters may be site-specific (i.e., more validation does not fully guarantee the proposed model works properly everywhere). After optimizing the parameters (i.e., $K_L$, $n$, and $r_0$), the parameters slightly changed from the default values (see Appendix C for

more details). Since the model results from before and after the parameter optimization were not statistically different in the error assessment, we still used the default values in a conservative way.

This method only considers the $E_{WC}$ from the canopy by neglecting the $E_{WC}$ from the trunk and stem. Besides, the interception of snow is not considered because the small amount of intercepted snowfall evaporates when the eddy covariance systems function improperly, and its melting and sublimation processes are much more complex than intercepted

rainfall. To distinguish snowfall from total precipitation, the empirical discriminants in Matsuo et al. (1981) were used. This method uses air temperature and humidity near the ground surface to separate snow from rainfall because when it snows, air is not saturated and the near ground air temperature is lower than that under rainy condition. The result from this method should be scrutinized by comparing it with other precipitation data, which are measured at a weather station near the site.

### 2.3.3 Gap-filling and partitioning technique for evapotranspiration: model-stats hybrid method

The currently used MDS is expected to under- and over-estimate ET under wet and dry canopy conditions, respectively due to the gap-filling without the consideration of canopy wetness (because the evaporative fraction is proportional to canopy wetness). Therefore, the gap-filling technique for ET proposed by Kang et al. (2012) was used: (1) to identify the canopy wetness, the intercepted canopy water ($W_c$, see Eq. 1) was calculated using the simplified Rutter sparse model; (2) all the missing gaps were filled by the MDS using the data under dry canopy conditions only (i.e., when $W_c = 0$), which corresponds

to the ET under dry canopy condition ($ET_{dry}$); (3) under wet canopy conditions (i.e., when $W_c > 0$), the gap-filled data were replaced with the sum of the $E_{WC}$ estimated by the simplified Rutter sparse model (i.e., $E_{WC\_Mod}$) and the $ET_{dry}$ multiplied by $1-(W_c/S)^n$ (i.e., the contribution from transpiration) (see Eqs. 1 and 2).

Such a gap-filled ET was partitioned into the transpiration ($T$ or ET from the dry canopy, which approaches the actual transpiration under a dense and closed canopy condition) and $E_{WC}$ as follows. In case of that the data was missing, the $T$ was

estimated as $(1-(W_c/S)^n)$ $ET_{dry}$, while the $E_{WC}$ was estimated as $E_{WC\_Mod}$. In case of that the data was not missing (i.e., $ET_{Obs}$), the partitioning procedure divided into two parts. If the signs of $ET_{Obs}$, $ET_{dry}$, and $E_{WC\_Mod}$ were the same, the $T$ was estimated by multiplying $ET_{Obs}$ and the ratios of $(1-(W_c/S)^n)$ $ET_{dry}$ to the sum of $(1-(W_c/S)^n)$ $ET_{dry}$ and $E_{WC\_Mod}$ (i.e., the estimated transpired-fraction of ET), while the $E_{WC}$ was estimated by multiplying $ET_{Obs}$ and the ratios of $E_{WC\_Mod}$ to the sum of $(1-(W_c/S)^n)$ $ET_{dry}$ and $E_{WC\_Mod}$ (i.e., the estimated evaporated-fraction of ET). If the signs of $ET_{Obs}$, $ET_{dry}$, and $E_{WC\_Mod}$

were not the same, the $T$ were estimated by $(1-(W_c/S)^n)$ $ET_{dry}$, while the $E_{WC}$ was estimated by subtracting $(1-(W_c/S)^n)$ $ET_{dry}$ (i.e., the estimated $T$) from $ET_{Obs}$. The procedure regarding the MSH is described in Fig. 1.

[Figure 1 here]

## 2.4 Correction and partitioning methods for the $CO_2$ flux

### 2.4.1 General nighttime $CO_2$ flux correction methods

The KoFlux protocol includes three different nighttime corrections methods: the friction velocity ($u^*$) filtering method (FVF), the light response curve method (LRC), and the modified van Gorsel method (VGF) (van Gorsel et al., 2009; Kang et al., 2014 and 2017). These three filtering methods each have their own way of selecting good quality $CO_2$ flux data. The site-specific settings of the individual methods were as follows: 1) the lower $u^*$ threshold for the FVF was 0.3 m s$^{-1}$ for both the GDK and GCK sites. (We checked the dependency of nighttime $CO_2$ flux on friction velocity during the growing and dormant seasons (not shown here). 0.3 m s$^{-1}$ of $u^*$ threshold can be determined during the growing season, while it is hard to clearly decide a threshold during the dormant season for the both sites. But, the threshold during the dormant season would be smaller than 0.3 m s$^{-1}$. Thus, we applied the constant threshold of 0.3 m s$^{-1}$ for the sites (conservative approach similarly to Reichstein et al., 2005)), 2) the Michaelis-Menten-type light response curve ( $NEE = R_{LRCd} - (\alpha Q_t A_{max} / \alpha Q_t + A_{max})$ , where $R_{LRCd}$ is the estimated mean daytime RE, $\alpha$ is the apparent quantum yield, $A_{max}$ is the canopy scale photosynthetic capacity, and $Q_t$ is the total incident shortwave radiation above the canopy; van Gorsel et al., 2009) was used for the LRC, and 3) the peak of the $F_{CO2\_Obs}$ that occurred approximately at sunset ($R_{max}$) was directly used for the modified VGF after calculating the median diurnal variation of the $CO_2$ flux for a certain period (Kang et al., 2014 and 2017).The modified VGF produces a similar result to that from the original VGF for the sites (see Appendix D for more details regarding the modified VGF). We applied a 30-day moving window to obtain the daily $R_{LRCd}$ and $R_{max}$.

The selected RE data from each filtering method were processed as follows. First, we estimated the parameters in the RE equation (Lloyd-Taylor equation, $RE = R_{ref} \exp(E_0(1/T_{ref} - T_0) - 1/(T_a - T_0)))$ , where $R_{ref}$ is the reference RE, $T_{ref}$ is the reference temperature (= 10°C), $E_0$ is the activation energy parameter (°C$^{-1}$), $T_0$ is –46.02°C and $T_a$ is the air temperature (°C), using the selected observed RE (Lloyd and Taylor, 1994). Second, we replaced the bad (or missing) data with the calculated data using the air temperature and the RE function with the estimated parameters. We estimated $R_{ref}$ using a 30-day moving window which was shifted every 5-days to consider the variations of a RE-controlled by soil moisture and phenology, which is not considered in the Lloyd-Taylor equation. The $E_0$ is constant for each site-year, which is estimated using the generic algorithm proposed by Reichstein et al. (2005) that derives a short-term temperature sensitivity (see Reichstein et al., 2005; Hong et al., 2009 for more details). Each method was independent each other. In cases of the LRC method and the modified VGF, the nighttime $CO_2$ fluxes were filtered out when the observed nighttime $CO_2$ fluxes were underestimated out of the 95% confidence interval of the RE model (i.e., Lloyd and Taylor equation). GPP was calculated by subtracting NEE from RE.

### 2.4.2 A modification of the moving point test method (MPT)

The objective of the moving point test (MPT) method is to determine the intermediate range of $u^*$ where the nighttime $CO_2$ fluxes are independent of $u^*$ (Gu et al., 2005). The method searches for lower and higher $u^*$ thresholds, which are found by statistically testing (i.e., $t$-test) a group of points with consecutive $u^*$ values in a narrow moving window against a reference sample. The original method excludes night data when the median $u^*$ were lower than the lower $u^*$ threshold, to avoid an underestimation of the $CO_2$ flux due to drainage flow. However, this consideration is inappropriate for hilly terrain sites that are usually affected by drainage flow (e.g., the study sites, GDK and GCK), resulting in that the $CO_2$ fluxes were close to 0 and/or much smaller than the true values during most of nighttime except near sunset.

Thus, we modified the original MPT method to apply it to hilly terrain sites by: (1) splitting it into the two time windows, i.e., the time window near sunset (when a drainage has not yet (completely) manifested. It had been directly/indirectly proven using the bulk Richardson number, the $CO_2$ concentration profile, the $CO_2$ flux measured by chamber method (van Gorsel et al., 2007, 2008) and the information flow between the uphill and downhill (Kang et al., 2017) in the previous studies.) and the time window after the former, (2) applying the MPT method to each time window, (3) comparing the results between the two time windows and determining the existence of $CO_2$ drainage (i.e., the averages of the normalized nighttime $CO_2$ fluxes for the two time windows were significantly different), and (4) excluding all the data in the second time window if the $CO_2$ drainage is commonly generated (i.e., showing the significant difference between the averages) and then applying the FVF method for the both time windows using the $u^*$ thresholds determined in the previous steps.

The best feature of the modified MPT is that the time is split into the two time windows based on van Gorsel et al. (2009): (1) for calculating the median diurnal variation of the $CO_2$ flux and identifying whether the peak of NEE occurred approximately at sunset; (2) splitting the time windows, i.e., the first window one and two hours before and after the time of peak occurrence, respectively; and the second window of the time after the first time window.

The assumption of this method is that the biological and meteoroidal conditions that can affect the RE are not significantly different between the two time windows. For the study sites, the mountain wind rose consistently at night, and we did not find any driver which makes a difference between the averages of RE in the two time windows except ecosystem temperature. Thus, we compared the averages after normalizing the flux measurements using the temperature response function (i.e., Lloyd and Taylor equation) as same as the original MPT method proposed by Gu et al. (2005).

For applying the MPT method, there were two criteria, i.e., the significance level in the $t$-test ($\alpha$) and the size (i.e., the number of data) of the moving sample ($n$). According to Gu et al. (2005), the $\alpha$ is 0.1, and the $n$ is 25. The MPT method was applied for every three months. The details regarding the (modified) MPT are described in the flow charts (Fig. 2) and Gu et al. (2005).

[Figure 2 here]

# 3 Results

## 3.1 Gap-filling and partitioning results for the H$_2$O flux

### 3.1.1 Validation of the MSH

First, we evaluated the latent heat flux under (mostly) wet canopy conditions ($\lambda ET_{WC}$, i.e., $\lambda ET$ when $W_c/S>2/3$) from the
model-stats hybrid (MSH) method ($\lambda ET_{WC\_MSH}$) against the observed $\lambda ET_{WC}$ ($\lambda ET_{WC\_Obs}$) at both sites from 2008 to 2010
(Fig. 3). Most of the points are near a one-to-one line. The data scattered away from the one-to-one line are characterized by
large aerodynamic conductance (e.g., >100 mm s$^{-1}$) and/or large VPD (e.g., >10 hPa).

[Figure 3 here]

Table 1 shows the statistical parameters for the error assessment (i.e., MBE, MAE, RMSE, $d$, slope, and $r^2$; see Appendix E
for more details about the error assessment). The slopes from the linear regression analysis are 0.97±0.15 and 0.89±0.07 with
0.69±0.06 and 0.81±0.02 of $r^2$ for the GDK and GCK sites, respectively. The $d$ values for the sites were close to 1
(0.91±0.01 for the GDK and 0.95±0.01 for the GCK). Compared to the previous research (i.e., Kang et al., 2012), the results
from the MSH were closer to the observation due to the consideration of ET from the dry canopy. One of the leading causes
of the error in $\lambda ET_{WC\_MSH}$ was identified as the discrepancy between the time when the rain occurred, and the tipping bucket
was tipped. To validate only $E_{WC}$, cross-validation using the other models (e.g., Gash sparse analytical model, Gash et al.,
1995) can be attempted (e.g., Kang et al., 2012). Overall, the results from the linear regression analysis of $\lambda ET_{WC\_MSH}$ and
$\lambda ET_{WC\_Obs}$ show that MSH can provide $\lambda ET_{WC}$ reasonably well for the sites.

[Table 1 here]

### 3.1.2 Comparison between the MDS and the MSH

To evaluate the superiority of the MSH, we filled up the missing $\lambda ET_{WC}$ data by using the MDS ($\lambda ET_{WC\_MDS}$) and the MSH
($\lambda ET_{WC\_MSH}$). The underestimation of the $\lambda ET_{WC\_MDS}$ had been shown by the comparison with the sum of energy flux
components except for latent heat flux (= net radiation + sensible heat flux + storage flux) in our previous study (Kang et al.
2012). The $\lambda E_{WC\_mod}$ displayed the mirrored patterns of the sum of the other energy budget components, while the $\lambda ET_{WC\_MDS}$
were very small (mainly due to the low radiation during the rainy days). Thus, we expected that the MDS underestimates the
ET since it cannot explicitly consider the key processes of wet canopy evaporation (i.e., the effects of aerodynamic
conductance ($g_a$) change and sensible heat advection, see Kang et al., 2012 for more detailed explanation). Actually, the
average annual MBEs from 2008 to 2010 were -18±6 W m$^{-2}$ for the GDK site and -15±5 W m$^{-2}$ for the GCK site,

respectively. It also should be noted that the $\lambda ET_{WC\_MSH}$ varied while $\lambda ET_{WC\_MDS}$ were nearly constant occasionally, because (1) the $\lambda ET_{WC\_Obs}$ rarely existed close to the missing data and (2) the MDS did not consider the effect of $g_a$ (not shown here). Figure 4 shows the monthly ETs gap-filled by the MDS and MSH methods for the GDK and GCK sites. First, the annual ETs from the MSH method were 16 ~ 41 mm year$^{-1}$, which is significantly larger than those from the MDS method, while the random uncertainties in gap-filled annual ETs were approximately 5 mm year$^{-1}$ for the both sites (quantified according to Finkelstein and Sims, 2001, and Richardson and Hollinger, 2007). The significant difference was identified in June, July, August, and September when it was intensive rainfall. The biggest difference is shown in 2010 with more frequent and larger rainfall (for the GDK, the number of rainy days is 86, 82, and 103 days and the total amount of rainfall is 1,407, 1,323, and 1,652 mm in 2008, 2009, and 2010, respectively. Such characteristics are similar to those for the GCK). In addition to taking the missing $E_{WC}$ from the MDS into account, the other advantage of the MSH method is that the observed in ET and by eddy covariance system can be partitioned into transpiration ($T$) and $E_{WC}$ without any additional measurement. However, it can be applied to a dense canopy only, where soil evaporation is negligible. Otherwise, (e.g., before leaf unfolding and after leaf fall), the $T$ includes the error of the soil evaporation ($E_S$). Thus, there is more separating the $E_{WC}$ than partitioning the ET. The annual $E_{WC}$ ranged from 53 mm to 82 mm for the GDK and 78 mm to 112 mm for the GCK, which occupies 14 ~ 23% and 14 ~ 19% of the annual ET, respectively.

For quantifying the $E_S$, the supplementary eddy covariance (EC) systems were operated at the floors of the GDK and GCK sites (Kang et al. 2009b). The annual understory ET (~ $E_S$) from 1 June 2008 to 31 May 2009 was 59 mm for the GDK and 43 mm for the GCK, which occupied 16% and 8% of the annual ET, respectively. The decoupling factor ($\Omega$, McNaughton and Jarvis, 1983) at the forest floor was ~ 0.15 for the both sites approximately, which indicates that the $E_S$ was controlled primarily by the VPD and surface conductance ($g_s$) rather than $R_{sdn}$. This factor also suggests that separating $E_S$ from ET using the exponential radiation extinction model to estimate the $R_{sdn}$ at the forest floor and the relationship between the estimated $R_{sdn}$ and the ET when the canopy is inactive (Stoy et al., 2006) can be problematic for the sites. Considering that the accurate estimation of $g_s$ is challenging, a supplementary measurement (e.g., low-level EC, lysimeter, sap-flow measurement, and isotope) is a better approach for estimating $E_S$. Using the $E_S$ measured by the low-level EC, the annual $T$ can be estimated at 265 mm (70% of ET) for the GDK and 448 mm (78% of ET) for the GCK, while the $E_{WC}$ estimated as 55 mm (15% of ET) and 82 mm (14% of ET), respectively (Fig. 4).

[Figure 4 here]

### 3.2 Correcting and partitioning results for the $CO_2$ flux

#### 3.2.1 Validation of the modified MPT

Figure 5 shows a part of the results from the modified moving point test (MPT) method for the GCK in 2008. During DOY 91-181, the averages of the normalized nighttime $CO_2$ fluxes (after the $u^*$ filtering) for the two time windows (i.e., the time window near sunset (when drainage has not yet (completely) manifested ($1^{st}$ time window) and the time window after the former ($2^{nd}$ time window)) was statistically the same. The result suggests that there is no drainage and/or the drainage effect is negligible. Meanwhile, during DOY 1-90, those are significantly different, suggesting that the observation for the $2^{nd}$ time window cannot represent the actual ecosystem respiration due to the drainage. Many points for the $2^{nd}$ time window are near 0 although the air temperature is higher than 0 °C (Fig. 5(b)). The results from the modified MPT are summarized in Table 2. It is hard to find a certain characteristic of the drainage effect, implying that it is a consequence complexly influenced by micrometeorology, phenology, and data availability. Such results require careful examinations from micrometeorological and ecological perspectives using the other independent observations because the modified MPT is a kind of empirical correction.

[Figure 5 here]

[Table 2 here]

To validate the results from the modified MPT, we compared the results to those from a previous study (Kang et al. 2017). They developed the site-specific quality control filter to exclude data strongly affected by $CO_2$ advection, which identifies the observations when a high information flow toward the bottom of the slope exists in the dynamical process network based on the observed multi-level $CO_2$ concentrations at the GDK and GCK sites. The filter discards the data (1) when the $CO_2$ drainage is fully generated as the mountain wind prevails (i.e., 21:00 to 9:00, 8:00, and 7:30 for the dormant, transition, and growing seasons, respectively), (2) when $u^*$ is lower than the threshold (0.3 m s$^{-1}$) while the drainage flow is under strong development (from 17:00 to 21:00), and (3) until the accumulated $CO_2$ completely dissipates for the downhill (GCK) site (i.e., before noon). Figure 6 shows the (averaged) annual $CO_2$ budget from the three general nighttime correction methods (see the Chapter 2.4.1) after applying the filter (Currently (without other independent measurements such as chamber method), it was the most reliable result) and that from the modified MPT method for the sites. Almost all the results agree with each other within the margin of error, indicating the validity of the modified MPT method. Additionally, we compared the annual net ecosystem production (NEP, = net primary production (NPP) – heterotrophic respiration) with the diameter at breast height (DBH) increment (which is directly related to the NPP) for the GCK site with the single dominant species (i.e., *Abies holophylla*). The both show similar interannual variability, which also can support the reasonability of the method (Fig. 7, even though there was a limitation because the heterotrophic respiration was not considered).

[Figure 6 here]

[Figure 7 here]

## 4 Applications

In this chapter, we illustrate the advantages of the proposed techniques. Mostly, the benefits are caused by the gap-filling and partitioning of the $H_2O$ flux because the model-stats hybrid (MSH) method can take the $E_{WC}$ into account properly and separate them from ET, which has not yet been previously possible. On the other hand, the modified moving point test (MPT) method has an improved applicability because it can be applied regardless of topography. We hope the following chapters draw attention to the ET partitioning.

### 4.1 Wavelet coherence analysis between ET and the rainfall

To evaluate the effect of new gap-filling, we conducted the wavelet coherence analysis between ET and rainfall for the GDK site (Fig. 8, see Hong et al. (2010) and Grinsted et al. (2004) for more details regarding the wavelet coherence analysis). From one- to the third month period, which was the three-month period in the monsoon season (i.e., the intensive rainy period), high correlation (i.e., red color area) was observed between ET and rainfall in 2006, 2007, 2008, and 2009. In 2007, the rainfall amount was 200 mm lower than the average level during the study period. However, the rainfall duration was the longest, and the intensity was the lowest. In 2006, 2008, and 2009, the arrow on the high correlation area pointed left. It means a negative correlation between the two variables, reflecting that the decrease in $T$ caused by the diminishment of $R_{sdn}$ during the intensive rainy period. In contrast, the arrow pointed right in 2007, indicating a positive correlation. The magnitude of enhanced $E_{WC}$ was greater than that of decreased $T$ at that time and frequency in 2007. Such a positive correlation between ET and rainfall with one- to three-month cycle in 2007 was not reported in the previous study of Hong et al. (2010) which showed a negative correlation in 2006, 2007 and 2008 at that time and frequency. This can be attributed to the improvement in ET data made by the new gap-filling method (i.e., recovering the missing $E_{WC}$ in the general gap-filling method). During the monsoon season, the $E_{WC}$ compensates (a portion of) the decreased $T$, and it can occasionally be balanced (e.g., in 2010).

There are some circumstantial evidences which support that the proposed method is more appropriate for taking $E_{WC}$ into account than the conventional method: (1) the ratio of the runoff and the precipitation (adapted from Choi et al. 2011) in 2007 was the lowest (0.60 in 2007, 0.69±0.06 in the other years, i.e., the ratio of the ET to the precipitation can be the highest in 2007), while the $R_{sdn}$ (main controlling factor of $T$) was the lowest (4.52 GJ m$^{-2}$ in 2007, 4.77±0.08 GJ m$^{-2}$ in the other years) due to the longest rainfall duration, (2) the interannual variabilities of the estimated catchment scale annual ET (i.e., precipitation – runoff) and ET from the MDS method occurred in opposite directions (similarly to $T$ from the MSH method).

[Figure 8 here]

## 4.2 Water use efficiency at the ecosystem-level and the canopy-level

Water use efficiency (WUE) can be defined in various forms such as $A_n/g_{st}$ (intrinsic WUE; $A_n$: net assimilation; $g_{st}$: stomatal conductance; $A_n$=NPP), $A_n/T$ (instantaneous WUE), $A_n(1-\Phi_c)/[T(1+\Phi_w)]$ (integrated WUE; $\Phi_c$: fraction of assimilated carbon lost in respiration; $\Phi_w$: fraction of total water loss from non-photosynthetic parts of the plant or through open stomata at night), the GPP/$T$ (canopy-level WUE), NPP/ET (stand-level WUE), and GPP/ET (ecosystem-level WUE) (Seibt et al., 2008; Ito and Inatomi, 2012; Ponton et al., 2006), because the spatiotemporal scale and measurement method are research-specific. Based on the original definition of WUE (i.e., the ratio of $CO_2$ flux to $H_2O$ flux), we re-defined the annual ecosystem-level WUE ($WUE_{Eco}$) and the annual canopy-level WUE ($WUE_{Canopy}$) as $\Sigma NEP/\Sigma ET$ and $\Sigma NPP/\Sigma T$, respectively. For estimating $\Sigma NPP$ and $\Sigma T$ simply, we used 0.45 of the ratio of the NPP to GPP for the both sites (Waring et al., 1998), and 0.156 and 0.075 of the ratios of $E_S$ to ET for the GDK and GCK, respectively (Kang et al., 2009b). From 2006 to 2010, $WUE_{Eco}$ ($WUE_{Canopy}$) ranged from -0.16 (2.17) to 0.32 (2.59) g C·(kg $H_2O)^{-1}$ for the GDK site and from 0.20 (1.93) to 0.38 (2.16) g C·(kg $H_2O)^{-1}$ for the GCK site (Table 3). Considering the increasing trend of NEE and GPP for the GCK site, it can be identified that the interannual variabilities of $WUE_{Eco}$ and $WUE_{Canopy}$ occurred in opposite directions for the both sites. It was primarily caused by that $E_{WC}$ were enhanced in 2007 and 2010 due to the weakest rainfall intensity and the largest rainfall amount, respectively. Overall, such partitioning of the total ET into $E_{WC}$, $T$, and $E_S$ enables us to understand better how ET responses to environmental changes and the how water cycle is connected to the carbon cycle in a forest ecosystem.

[Table 3 here]

## 5 Conclusions

There is a common characteristic between the two techniques proposed in this study for gap-filling and partitioning of $H_2O$ and $CO_2$ eddy fluxes: two existing methods were merged into a new method. The marginal distribution sampling (MDS) method and the simplified Rutter spars model have merged into the model-stats hybrid (MSH) method and the $u^*$ filtering (i.e., moving point test method, MPT) and van Gorsel methods were merged into the modified MPT. Such a strategy strengthens the strength and makes up for the weakness of the original methods. Especially, the modified MPT for nighttime $CO_2$ flux correction substantially improves its applicability, expecting that it will contribute to the standardization of eddy covariance data processing. In this context, such attempt will and must continue. The MSH method can be applied to tropical forests because tropical forests also share three properties of temperate forests (i.e., extensive, dense, and tall). However, applying the methods to grasslands may need further validation.

**Acknowledgments**

This work was supported by the Korea Meteorological Administration Research and Development Program under Grant KMIPA 2015-2023, and by the Weather Information Service Engine (WISE) project, KMA-2012-0001-2. We thank Hyojung Kwon, Jinkyu Hong, Jaeill Yoo, Juyeol Yun, and Je-woo Hong for their helpful support in the data collection and other logistics. Wavelet tools used in this study are benefited from Grinsted et al. (2004).

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

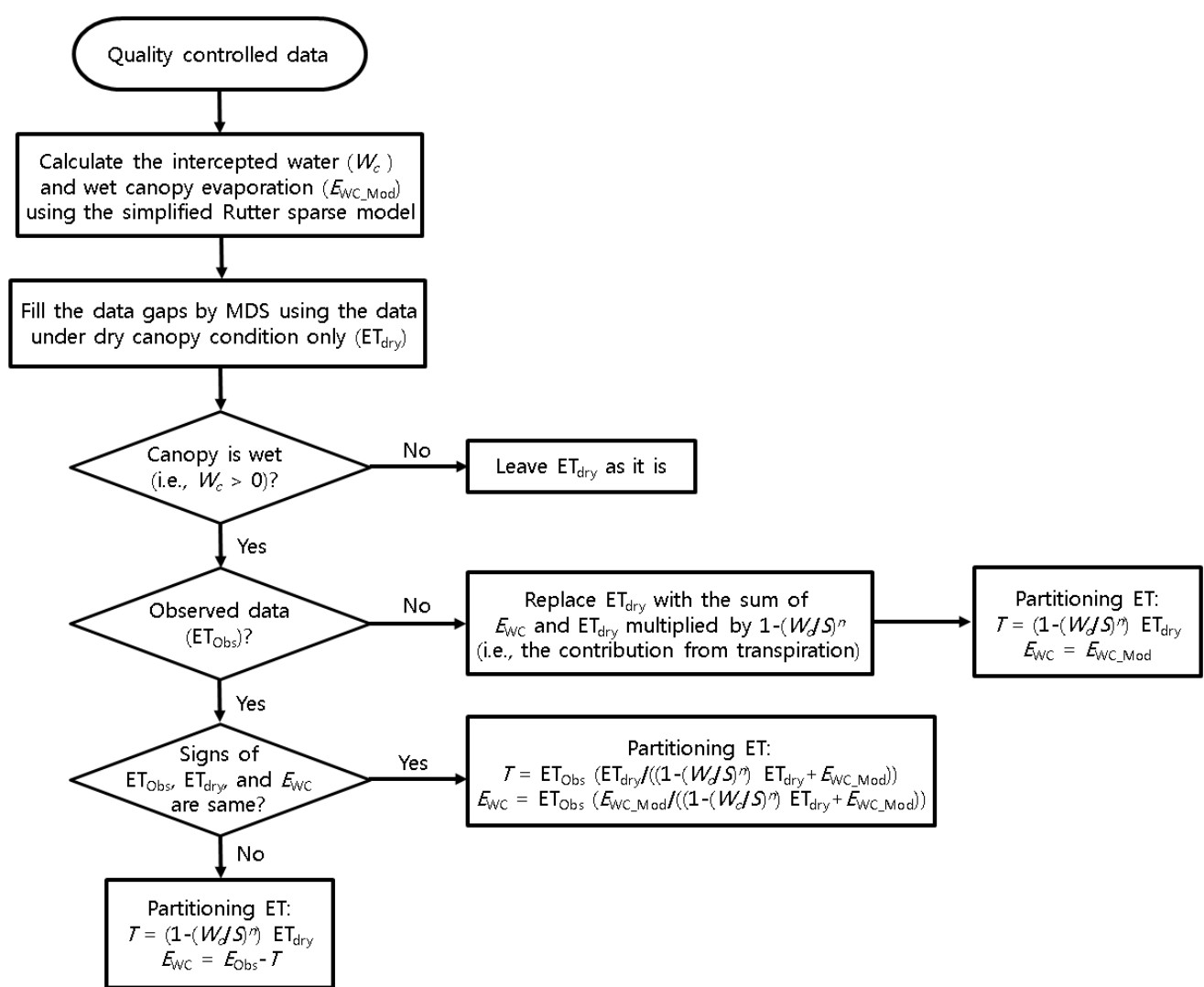

**Figure 1: Flowchart of the gap-filling and partitioning technique for evapotranspiration.**

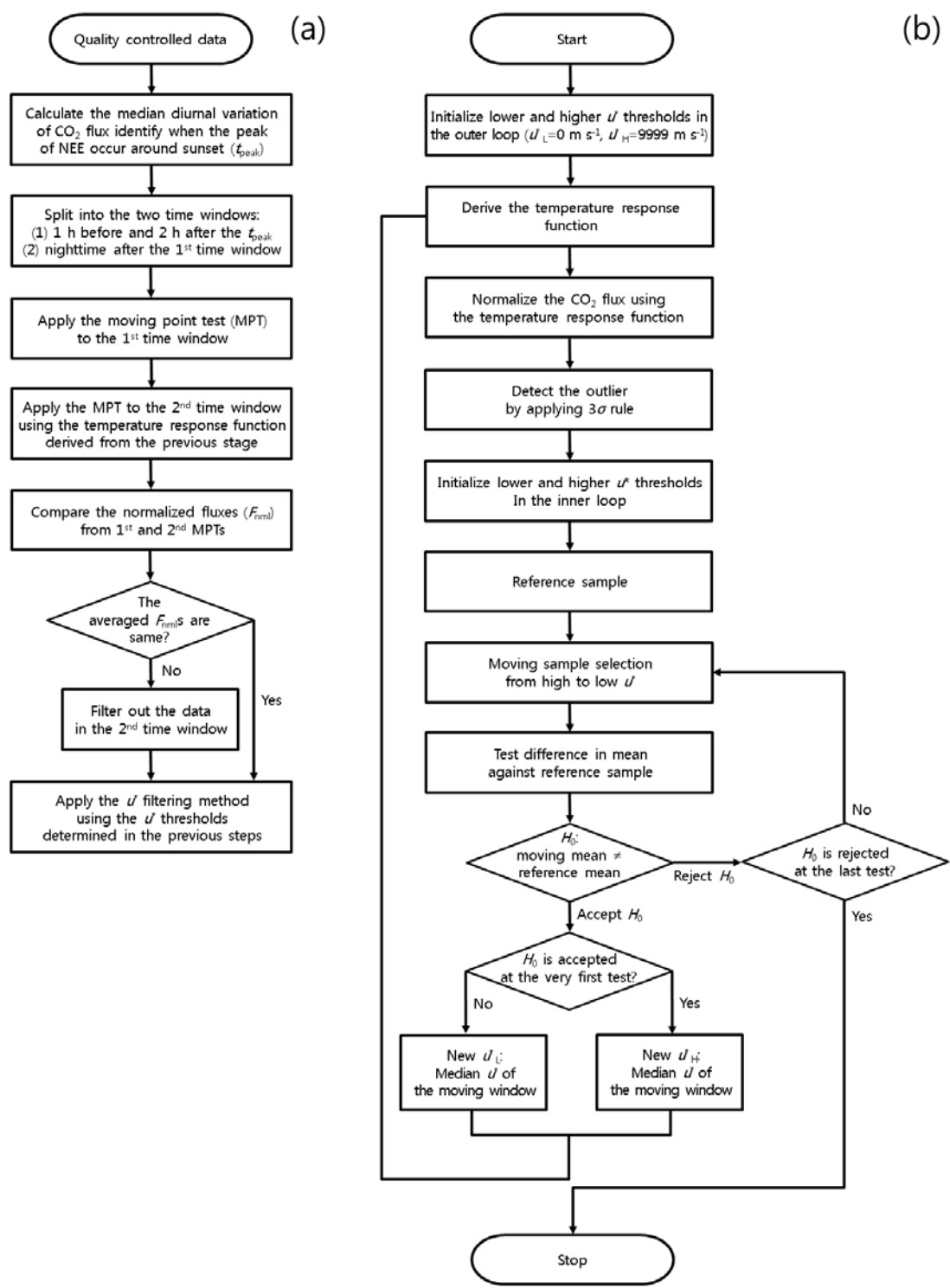

**Figure 2: Flowcharts of the gap-filling and partitioning technique for $CO_2$ flux, the modified moving point test (MPT) method (a) and the original MPT (b; adapted from Gu et al. 2005).**

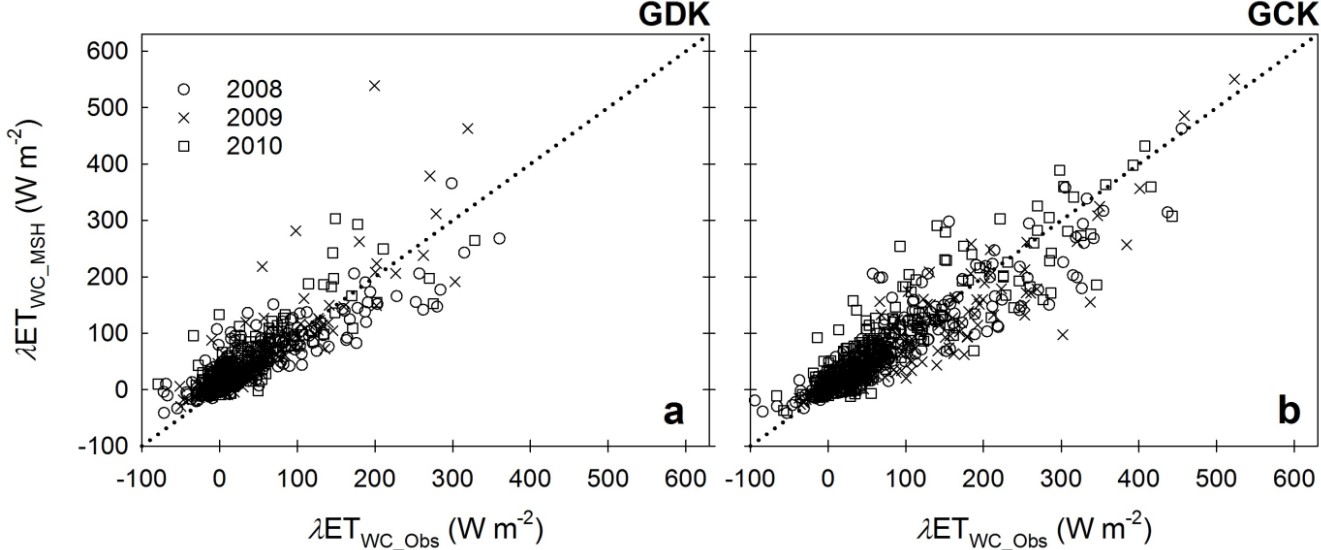

**Figure 3: Comparison of the latent heat flux under (mostly) wet canopy condition (i.e., $W_c/S>2/3$ where $W_c$ is the intercepted canopy water and $S$ is the canopy storage capacity) at the GDK (a) and GCK (b) sites: $\lambda ET_{WC\_Obs}$ indicates the observed latent heat flux under a wet canopy condition ($\lambda ET_{WC}$), while $\lambda ET_{WC\_MSH}$ indicates the estimated $\lambda ET_{WC}$ from the model-stats hybrid method.**
5    **The dotted line represents the 1:1 line.**

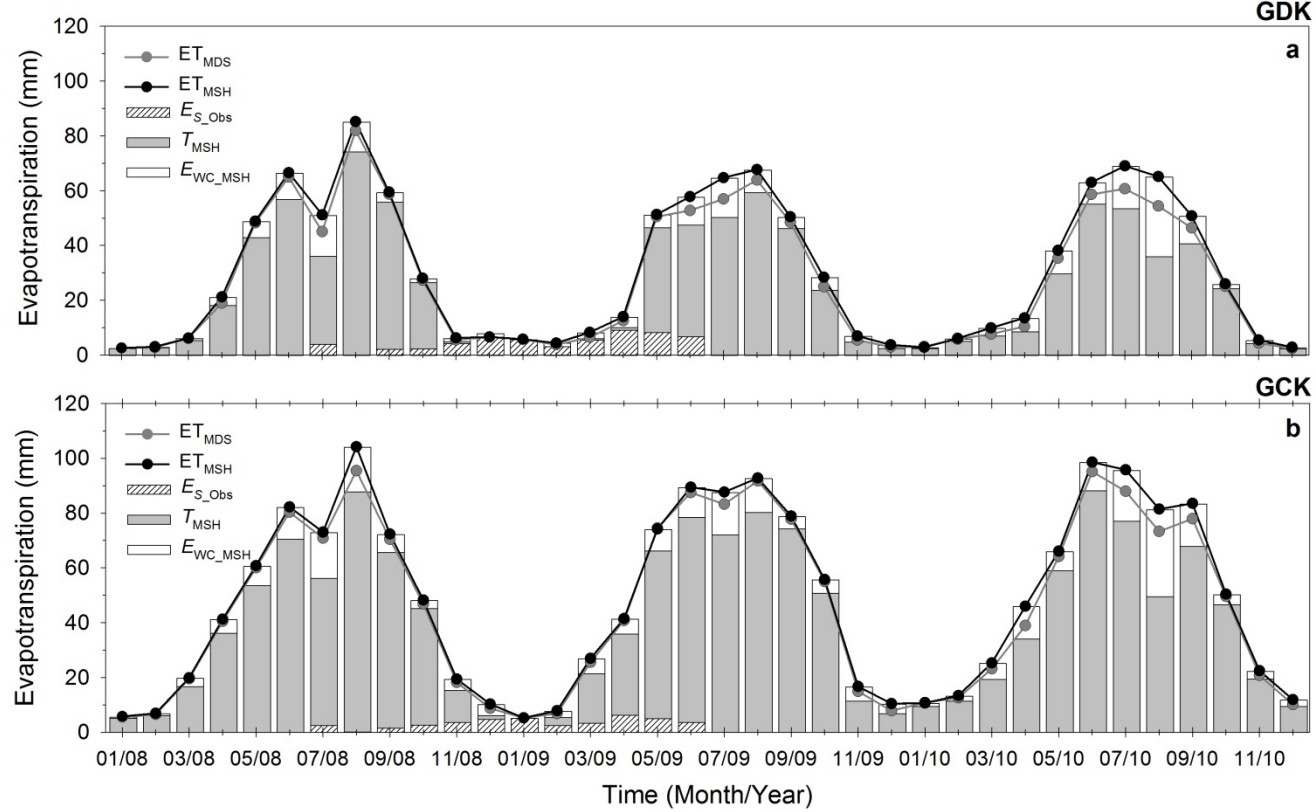

**Figure 4: Seasonal variation of monthly integrated evapotranspiration (ET) with the gap-filled by the marginal distribution sampling method (ET$_{MDS}$); the ET gap-filled by the model-stats hybrid (MSH) method (ET$_{MSH}$), transpiration and wet canopy evaporation partitioned by the MSH method ($T_{MSH}$ and $E_{WC\_MSH}$), for the GDK (a) and GCK (b) sites. $E_{S\_Obs}$ indicates soil evaporation measured by the supplementary eddy covariance systems at the floors (adapted from Kang et al., 2009b).**

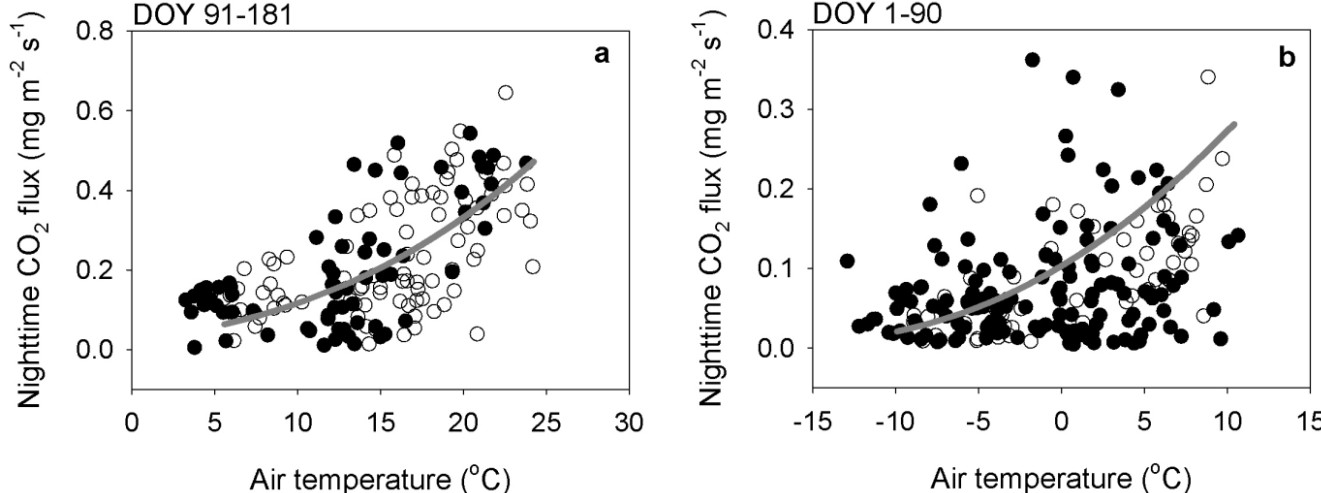

**Figure 5: The relationship between the nighttime CO₂ flux and air temperature after applying the modified moving point test (MPT) (filtered by the $u^*$ thresholds which are determined from the modified MPT) for the GCK in 2008, DOY 91-181 (a) and DOY 1-90 (b). The white color indicates the 1ˢᵗ time window, whereas black color indicates the 2ⁿᵈ time window. The solid line indicates the fitting line of Lloyd-Taylor equation using the filtered data of the 1ˢᵗ time window.**

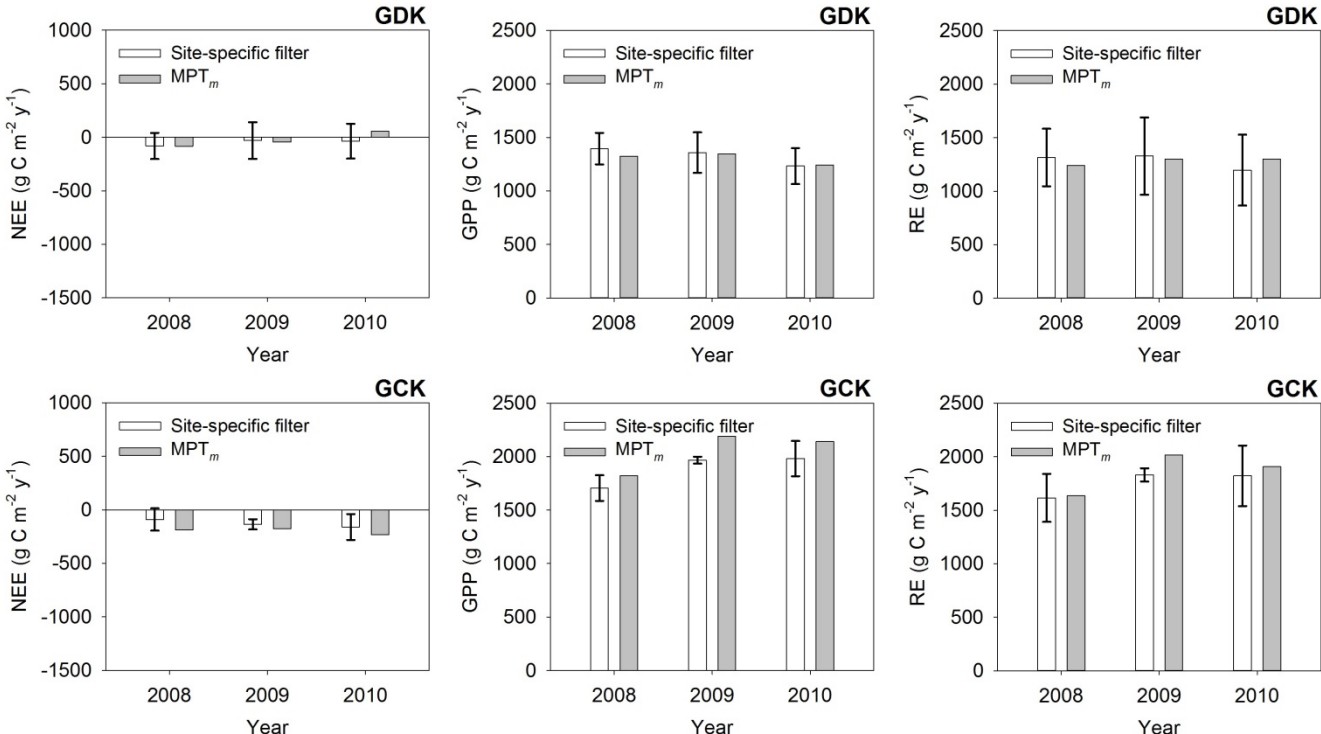

**Figure 6: The (averaged) annual $CO_2$ budget (NEE: net ecosystem exchange, GPP: gross primary production, RE: ecosystem respiration) from the three general nighttime correction methods (i.e., $u^*$ filtering method, light response curve method, and van Gorsel method) after applying the site-specific filter (adapted from Kang et al., 2017) and that from the modified MPT method for the sites. Error bar indicates the standard deviation of the results from the three general nighttime correction methods.**

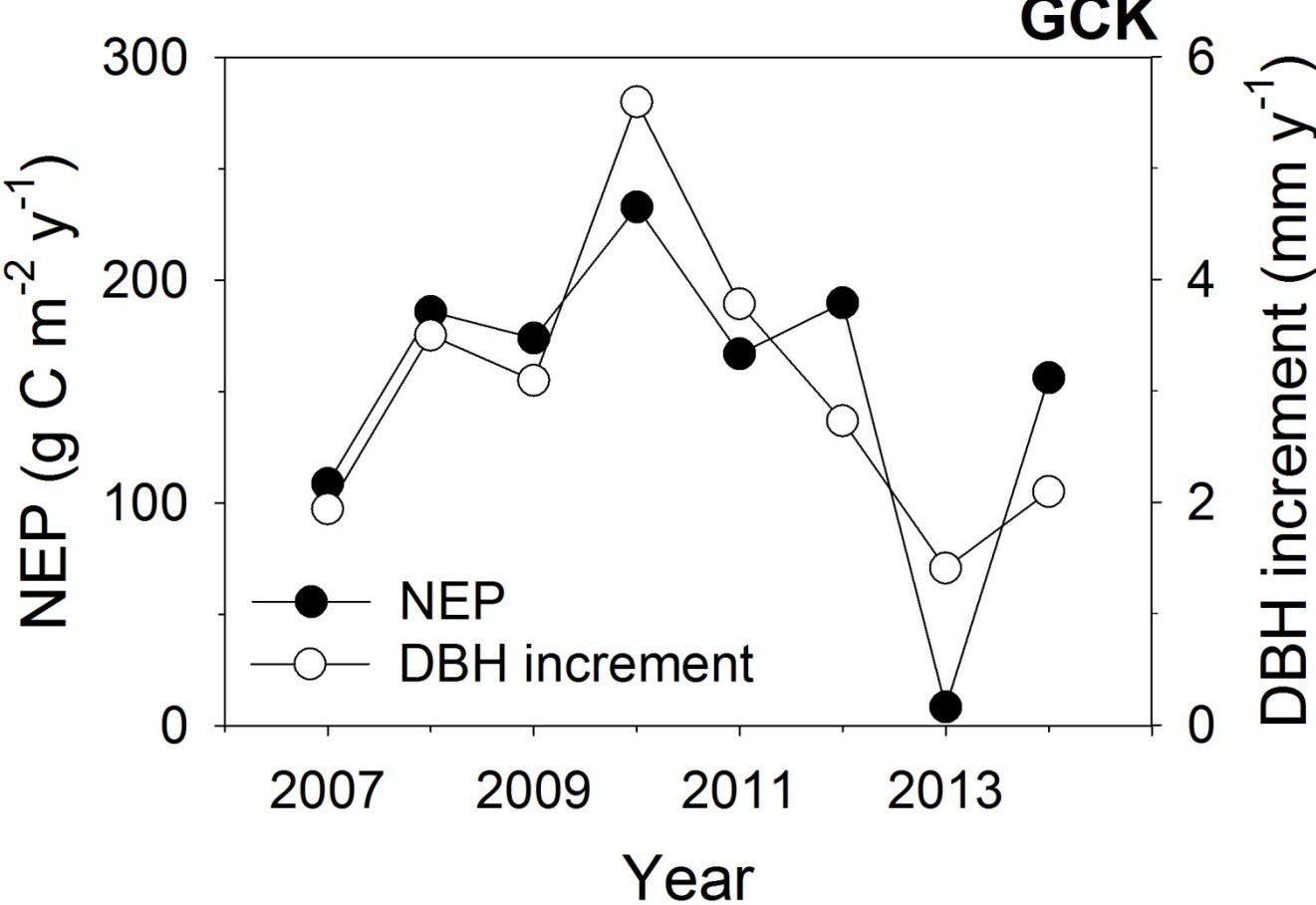

**Figure 7: The interannual variabilities of the annual net ecosystem production and the annual increment of DBH (diameter at breast height) for the GCK site.**

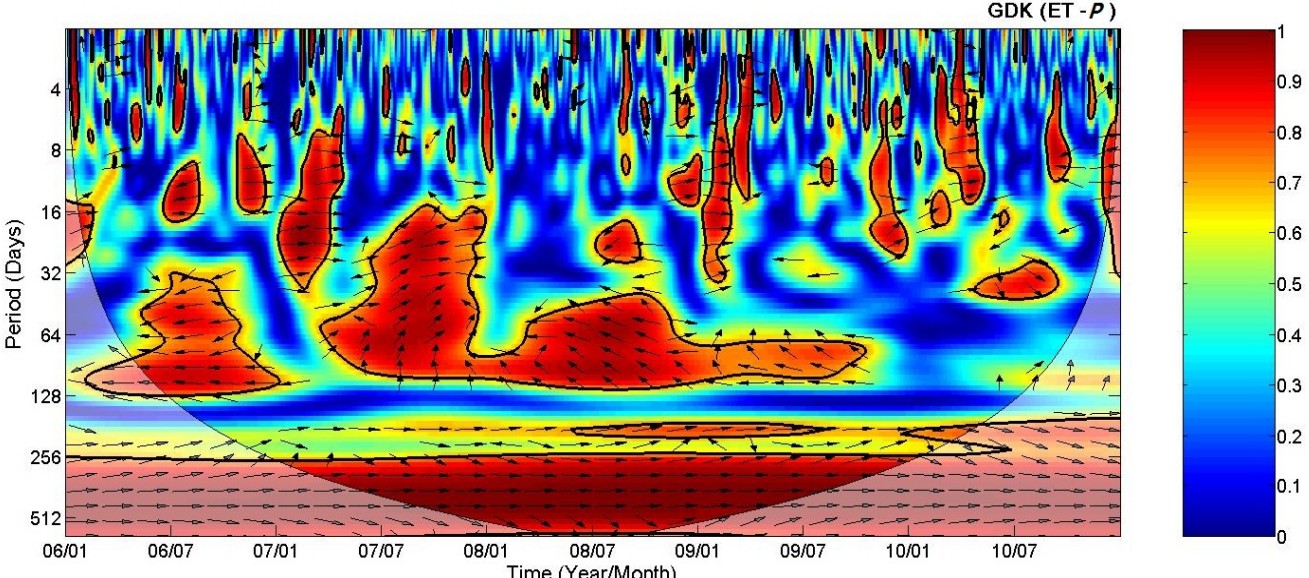

**Figure 8: Wavelet coherence spectrum of evapotranspiration (ET) with rainfall (*P*) for the GDK site. A thick solid contour is the 5% significance level against red noise as calculated from a Monte Carlo simulation. Arrows are the relative phase angle (with in-phase (positive correlation) pointing right, antiphase (negative correlation) pointing left, and *P* leading ET by 90° pointing straight down). The shaded area indicates the cone of influence where the edge effects might distort the results.**

**Table 1: Statistical parameters for the error assessment at the study sites. MBE, MAE, RMSE, and *d* indicate mean bias error, mean absolute error, root mean square error, and index of agreement, respectively. Slope and $r^2$ are from the linear regression analysis.**

|  |  | No. of data | MBE | MAE | RMSE | $d$ | Slope | $r^2$ |
|---|---|---|---|---|---|---|---|---|
|  |  | -- | W m$^{-2}$ | W m$^{-2}$ | W m$^{-2}$ | -- | -- | -- |
| GDK | 2008 | 333 | 6 | 20 | 30 | 0.93 | 0.80 | 0.72 |
|  | 2009 | 222 | 12 | 21 | 39 | 0.91 | 1.10 | 0.73 |
|  | 2010 | 215 | 14 | 23 | 34 | 0.90 | 1.00 | 0.63 |
| GCK | 2008 | 318 | -4 | 23 | 36 | 0.95 | 0.84 | 0.83 |
|  | 2009 | 246 | -10 | 26 | 44 | 0.94 | 0.85 | 0.79 |
|  | 2010 | 285 | 7 | 24 | 39 | 0.95 | 0.97 | 0.82 |

**Table 2: Summary results generated from the modified moving point test (MPT) for the GDK and GCK sites. $u^*_L$ and $u^*_H$ indicate lower and higher $u^*$ thresholds, $1^{st}$ and $2^{nd}$ indicate the $1^{st}$ and $2^{nd}$ time windows, respectively.**

| Site | Year | DOY | $u^*_L - 1^{st}$ | $u^*_H - 1^{st}$ | $u^*_L - 2^{nd}$ | $u^*_H - 2^{nd}$ | Drainage effect |
|------|------|-----|------|------|------|------|------|
| GDK | 2008 | 1-90 | 0.179 | 9999 | 0.337 | 9999 | negligible |
| | | 91-181 | 0.215 | 9999 | 0.363 | 9999 | significant |
| | | 182-273 | 0.178 | 9999 | 0.215 | 9999 | negligible |
| | | 274-366 | 0.144 | 0.271 | 0.171 | 9999 | negligible |
| | 2009 | 1-90 | 0.216 | 9999 | 0.411 | 9999 | negligible |
| | | 91-181 | 0.406 | 9999 | 0.169 | 9999 | significant |
| | | 182-273 | 0.238 | 9999 | 0.156 | 9999 | significant |
| | | 274-365 | 0.338 | 9999 | 0.184 | 0.506 | significant |
| | 2010 | 1-90 | 0 | 9999 | 0.326 | 0.47 | negligible |
| | | 91-181 | 0 | 9999 | 0.202 | 0.341 | significant |
| | | 182-273 | 0.247 | 9999 | 0.173 | 9999 | negligible |
| | | 274-365 | 0.166 | 9999 | 0.255 | 9999 | negligible |
| GCK | 2008 | 1-90 | 0.409 | 9999 | 0.174 | 9999 | significant |
| | | 91-181 | 0.29 | 9999 | 0.25 | 9999 | negligible |
| | | 182-273 | 0.164 | 9999 | 0.221 | 9999 | negligible |
| | | 274-366 | 0.149 | 9999 | 0.197 | 9999 | negligible |
| | 2009 | 1-90 | 0.248 | 9999 | 0.425 | 9999 | negligible |
| | | 91-181 | 0.252 | 9999 | 0.122 | 9999 | significant |
| | | 182-273 | 0.171 | 9999 | 0.141 | 0.198 | significant |
| | | 274-365 | 0.232 | 9999 | 0.238 | 9999 | negligible |
| | 2010 | 1-90 | 0.157 | 9999 | 0.295 | 9999 | significant |
| | | 91-181 | 0.157 | 9999 | 0.137 | 9999 | significant |
| | | 182-273 | 0.125 | 9999 | 0.122 | 0.216 | significant |
| | | 274-365 | 0.098 | 0.417 | 0.348 | 9999 | significant |

**Table 3: The annual $CO_2$ and $H_2O$ budget (NEE: net ecosystem exchange, GPP: gross primary production, RE: ecosystem respiration, ET: evapotranspiration, $E_{WC}$: wet canopy evaporation) and water use efficiency at the ecosystem-level ($WUE_{Eco}$) and the canopy-level ($WUE_{Canopy}$) for the study sites.**

| | | NEE | GPP | RE | ET | $E_{WC}$ | $WUE_{Eco}$ | $WUE_{Canopy}$ |
|---|---|---|---|---|---|---|---|---|
| | | g C m$^{-2}$ y$^{-1}$ | g C m$^{-2}$ y$^{-1}$ | g C m$^{-2}$ y$^{-1}$ | mm | mm | g C·(kg H$_2$O)$^{-1}$ | g C·(kg H$_2$O)$^{-1}$ |
| GDK | 2006 | -114 | 1,149 | 1,035 | 361 | 66 | 0.32 | 2.17 |
| | 2007 | -14 | 1,183 | 1,169 | 398 | 116 | 0.03 | 2.42 |
| | 2008 | -84 | 1,326 | 1,242 | 383 | 53 | 0.22 | 2.20 |
| | 2009 | -45 | 1,346 | 1,301 | 360 | 56 | 0.12 | 2.45 |
| | 2010 | 58 | 1,242 | 1,300 | 353 | 82 | -0.16 | 2.59 |
| GCK | 2007 | -109 | 1,892 | 1,783 | 557 | 122 | 0.20 | 2.16 |
| | 2008 | -186 | 1,822 | 1,636 | 544 | 78 | 0.34 | 1.93 |
| | 2009 | -174 | 2,190 | 2,016 | 587 | 77 | 0.30 | 2.12 |
| | 2010 | -233 | 2,140 | 1,907 | 606 | 112 | 0.38 | 2.15 |

**Appendix A: The differences in annual budgets of $CO_2$, latent, and sensible heat fluxes depending on the estimations of storage terms**

To estimate net ecosystem exchange (NEE) for a target ecosystem using eddy covariance technique, particularly for forests, we should consider not only the eddy flux term but also the storage term, i.e., the change of stored mass and energy of the air in the control volume. Particularly, the storage terms make a large contribution to total NEE under calm condition (i.e., weak turbulence condition). The storage term of carbon dioxide ($CO_2$) was calculated as follows (e.g., Aubinet et al., 2001; Papale et al., 2006):

$$S_c = \int_0^h \frac{P_a(z)}{R \cdot T_a(z)} \frac{\partial c(z)}{\partial t} dz$$

(A1)

where $S_c$ is the storage term of $CO_2$ ($\mu$mol m$^{-2}$ s$^{-1}$), $P_a$ is the barometric pressure (kPa), $R$ is the gas constant ($8.314 \times 10^{-3}$ kPa m$^3$ K$^{-1}$ mol$^{-1}$), $c$ is the $CO_2$ concentration ($\mu$mol mol$^{-1}$), $z$ is height (m), $t$ is time (s), and $h$ is the height of the control volume (usually, the height of the eddy covariance system). The storage terms of water vapor ($H_2O$) and heat are also calculated similarly to Eq. (4). The eight-level vertical profile systems for $CO_2$, $H_2O$, and air temperature are operated at the GDK and GCK sites (Yoo et al., 2009). The storage terms are calculated by finite difference method using the data from profile system. If profile system is not installed or is not operated normally, the storage terms are calculated using the data from eddy covariance system (i.e., considering single-level scalar variation only) under the assumption that the concentrations and air temperature are constant with height (Papale et al. 2006). Such calculated storage terms of $CO_2$, $H_2O$, and air temperature are added to those eddy covariance terms for estimating those NEEs.

Table A1 shows the differences in annual budgets of $CO_2$ flux / evapotranspiration (ET) / sensible heat flux ($H$) among the estimations of storage terms for the study sites in 2008. The most notable result is that the $CO_2$ budgets, i.e., the NEE, gross primary production (GPP), and ecosystem respiration (RE) show the large differences among the estimations of storage terms. Meanwhile, the ET and $H$ budgets are approximately identical without regard to the estimations of storage terms. The main reason of the minor effect of the storage term estimation in ET and $H$ is that the fluxes are always very low at night, when the storage term is important. As the other possible causes, $CO_2$ is heavier than air, and significant source and sink exist at various levels (e.g., soil, canopy) that are unlikely ET and $H$. For the $CO_2$ budgets, the more correctly the storage terms were estimated, the larger the NEE (i.e., stronger carbon sink), GPP, and RE are quantified primarily due to the underestimation of nighttime $CO_2$ fluxes (i.e., RE). Especially if the storage terms are not considered, there were large differences compared to the cases of considering the storage terms using any approach. Such results suggest that we should measure and process more systematically and accurately not only the eddy flux term but also the storage term for quantifying NEE for the forest ecosystem.

**Table A1: Annually integrated net ecosystem exchange (NEE), gross primary production (GPP), and ecosystem respiration (RE), evapotranspiration (ET), and sensible heat flux (*H*) depending on the estimation of the storage term (profile: estimate of storage term using profile data, single: estimation of storage term using one level (top) data, no storage: not considering storage term; if profile data was not available, we also excluded eddy covariance data for a fair comparison. Accordingly, the values could be slightly different to those in Table 3. MDS and MPT$_m$ mean marginal distribution sampling and modified moving point test methods, respectively.**

| | | NEE | GPP | RE | ET | *H* |
|---|---|---|---|---|---|---|
| | | | g C m$^{-2}$ y$^{-1}$ | | mm y$^{-1}$ | MJ m$^{-2}$ y$^{-1}$ |
| | | | MDS & MPT$_m$ | | | MDS |
| | Profile | -129 | 1,297 | 1,169 | 367 | 605 |
| GDK | Single | -181 | 1,209 | 1,028 | 365 | 604 |
| | No storage | -266 | 940 | 674 | 363 | 605 |
| | Profile | -147 | 1,614 | 1,467 | 533 | 1,085 |
| GCK | Single | -224 | 1,539 | 1,315 | 533 | 1,085 |
| | No storage | -290 | 1,280 | 990 | 527 | 1,092 |

## Appendix B: Parameterizations of canopy storage

The rainfall interception is sensitive to the change of canopy storage capacity ($S$) and vegetation fraction ($\sigma_f$ i.e., 1-gap fraction) (e.g., Shi et al., 2010). One of the characteristics common among the temperate forests considered in this study is a dense canopy. It means that the vegetation fractions in the forests are close to 1 (except before leaf unfolding and after leaf fall periods). Moreover, the gap fraction can be measured using a plant canopy analyzer with relative ease. Therefore, the error of the result from the model is mostly derived from the parameterization of $S$ (see Appendix C for more detailed information). The canopy storage capacity is affected by not only leaf area but also the other factors such as leaf shape, leaf angle, leaf/shoot clumping and hydrophobicity (water repellency) of a leaf (e.g., Crockford and Richardson, 2000). Additionally, the relationships between $S$ and these characteristics are changeable according to meteorological conditions (e.g., the wind, rainfall intensity), which make the parameterization of $S$ difficult (e.g., Dunkerley, 2009). Therefore, we used the simple parameterization of $S$ of VIC LSM (i.e., $S = K_L \times$ leaf (or plant) area index, where $K_L = 0.2$; Liang et al., 1994) and evaluated whether the parameterization was reasonable or not. The relationships between the leaf area index (LAI) and $S$ in the previous studies are presented in Fig. B1, indicating that the parameterization in VIC LSM (i.e., $K_L = 0.2$) is reasonable and the $K_L$ ranges from 0.1 to 0.3. Further studies on the parameterization of $S$ using leaf structure (e.g., leaf shape, leaf angle, leaf/shoot clumping) would be worth conducting for more accurate estimation of wet canopy evaporation.

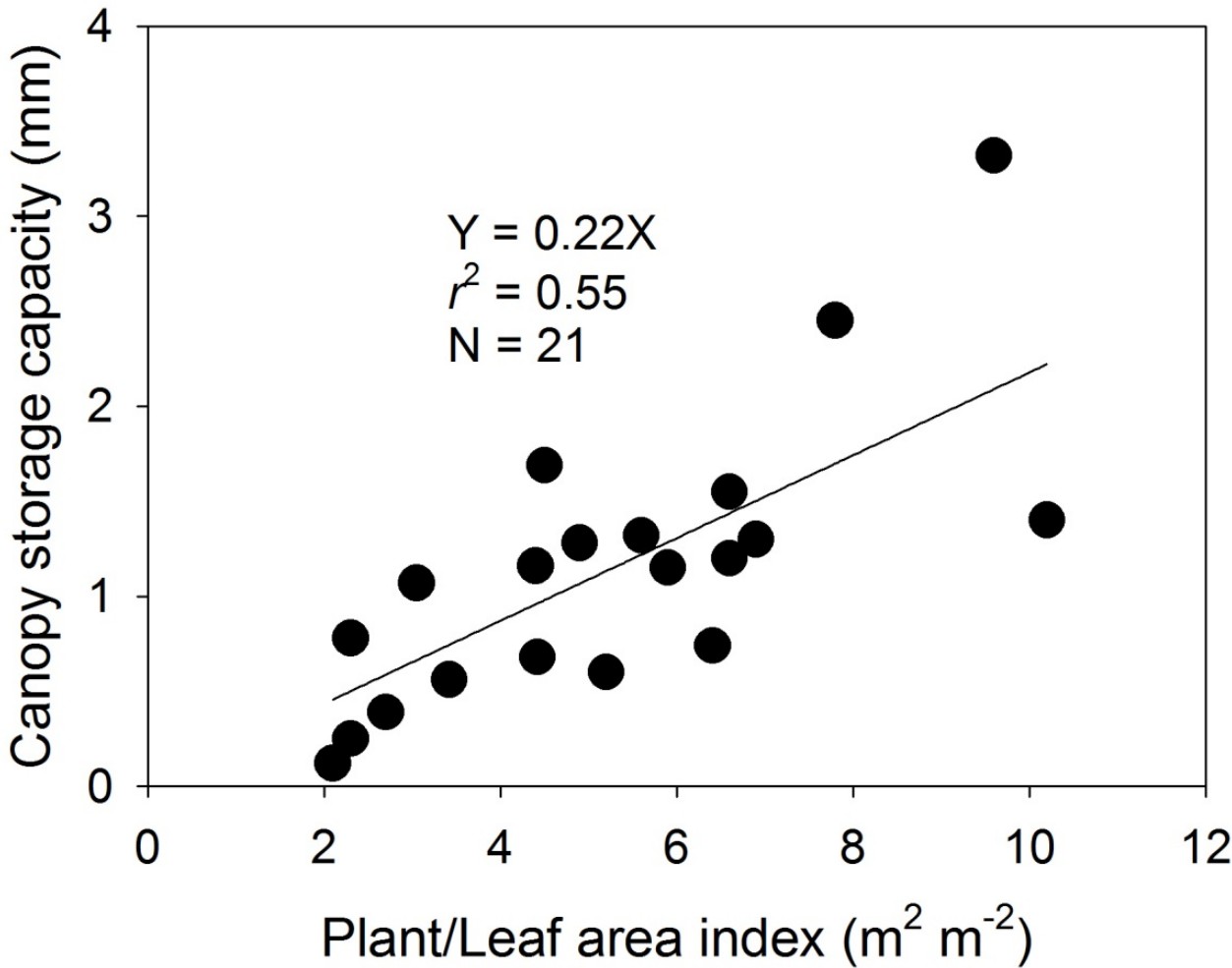

Figure B1: Relationship between canopy storage capacity and plant/leaf area index (the data obtained from Table B1).

**Table B1: Review of the canopy storage capacity for wet canopy evaporation modeling in the previous studies.**

| Vegetation type | Species | Country | Longitude /Latitude | Canopy storage capacity (mm) | Plant/Leaf area index ($m^2$ $m^{-2}$) | Density (trees ha$^{-1}$) | References |
|---|---|---|---|---|---|---|---|
| Laurel forest | *Myrica faya Ait., Laurus azorica(Seub.) Franco, Persea indica (L.) Spreng, Ericaarborea L., Ilex perado ssp. plathyphylla Webb & Amarra, Ilex canariensis Poivet* | Spain | 28°27′N, 16°24′W | 2.45 | 7.8 | 1,693 | Aboal et al. (1999) |
| Pine-oak forests | *Pinus pseudostrobus, Q.canbyi, Q.laeta* | Mexico | 24°42.3′N, 99°52.2′W | 0.78 | 2.3 | 819 | Carlyle-Moses and Price (2007) |
| Mixed agricultural cropping system | *Manihot esculenta Crantz, Zeamays L., Oryza sativa L.* | Indonesia | 7°03′S, 108°04′W | 0.12 | 2.1 | | van Dijk and Bruijnzeel (2001) |
| Plantation forest of Maritime pine | *Pinus pinaster Ait.* | France | 44°5′N, 0°5′W | 0.25 | 2.3 | 430 | Gash et al. (1995) |
| Norway spruce and Scots pine | *Picea abies (L.) Karst., Pinus sylvestris (L.)* | Sweden | 60°5′N, 17°29′E | 1.69 | 4.5 | | Lankreijer et al. (1999) |
| Secondary broad leaved deciduous forests | *Quercus serrara, Clethra barbinervis* | Japan | 35°02′N, 137°11′E | 1.07 | 3.05 | | Deguchi et al. (2006) |
| Mature rain forests | | Colombian Amazonia | | 1.16 | 4.4 | | Marin et al. (2000) |
| | | | | 1.28 | 4.9 | | |
| | | | | 1.32 | 5.6 | | |
| | | | | 1.55 | 6.6 | | |
| Tabonuco type forest | *Dacryodes excelsa* | Puerto Rico | 18°18N, 65°5′W | 1.15 | 5.9 | | Schellekens et al. (1999) |
| Mixed White Oak Forest | *Quercus serrata Thunb., Sasa paniculata Makino et Shibata.* | Japan | 35°19′N, 133°35′E | 0.6 | 5.2 | | Silva and Okumura (1996) |
| Secondary broad-leaved deciduous forests(Summer) | | Japan | | 0.68 | 4.42 | | Park (2000) |
| Secondary broad-leaved | | | | 0.39 | 2.7 | | |

| | | | | | | |
|---|---|---|---|---|---|---|
| deciduous forests(Winter) | | | | | | |
| Secondary broad-leaved deciduous forests (Summer) | | | | 0.74 | 6.41 | |
| Secondary broad-leaved deciduous forests (Winter) | | | | 0.39 | 3.42 | |
| Douglas-fir forest (Young) | | USA | 45°49.1′N, 121°59.7′W | 1.4 | 10.2 | Pypker et al. (2005) |
| Douglas-fir forest (Old) | | | 45°49.2′N, 121°54.1′W | 3.32 | 9.6 | |
| Decidous mixed forest (South) | *Carpinusorientalis croaticus, Quercus pubescentis* | Slovenia | | 1.2 | 6.6 | Sraj et al., (2008) |
| Deciduous mixed forest (North) | *F. ornus, Q. pubescentis* | | | 1.3 | 6.9 | |

**Appendix C: Sensitivity test and parameter optimization of the wet canopy evaporation model**

In the simplified Rutter sparse model (Liang et al., 1994), there are many parameters (e.g., $\sigma_f$, $S$, $n$, $r_a$, and $r_0$) for estimating the wet canopy evaporation ($E_{WC}$) and the intercepted canopy water ($W_c$). Since the model results may be sensitive to the parameters and the parameters may be site-specific, the parameter optimization using available flux data under wet canopy condition should be accompanied for the generalization of the model. Considering that the gap-filling and partitioning are a kind of interpolation and extrapolation (i.e., identifying relationships between a target flux and its drivers, and interpolating and extrapolating the relationships), it is an appropriate strategy for the gap-filling and partitioning of evapotranspiration using the model.

First, we conducted a sensitivity test of the model to the parameters (i.e., $k$, $K_L$, $n$, and $r_0$) using the dataset in 2008

$$\text{Change in } E_{WC}\ (\%) = \frac{E_{WC\_perturb} - E_{WC\_default}}{E_{WC\_default}} \times 100\ ,$$

$E_{WC\_default}$: Annually integrated $E_{WC}$ simulated with default parameters, $E_{WC\_perturb}$: Annually integrated $E_{WC}$ simulated after a change in each parameter. Only one parameter is changed one at a time, while other parameters are held in constants, e.g., Shi et al., 2010). Before testing the sensitivity, we set the lower/upper boundaries (and default values) based on the literature reviews: $k = 0.3 \sim 1.5$ (Jones, 2013; The default values of $k$ are 0.75 and 0.485 for the GDK and GCK, respectively. Those values were obtained from the actual measurement using a plant canopy analyzer (Model LAI-2000; Li-Cor Inc.)); $K_L = 0.1 \sim 0.3$ (see Appendix B; The default values of $K_L$ is 0.2 (Dickinson, 1984).); $n = 0.5 \sim 1$ (Chen and Dudhia, 2001; Liang et al., 1994; Valente et al., 1997; The default values of $n$ is 2/3 (Deardorff, 1978).); $r_0 = 0$ (for short vegetation) $\sim 2$ s m$^{-1}$ (for tall vegetation) (Perrier, 1975; Rana et al., 1993; The default values of $r_0$ is 2 (Perrier, 1975).). $K_L$ is the most influential parameters (Fig. C1), implying that we should take great care to minimize parameter estimation error for $K_L$.

Using a small number of the observed latent heat flux data under wet canopy condition (when $W_c/S > 2/3$) from 2008 to 2010, we optimized the parameters except $k$ (because we obtained the $k$ from the actual measurement) towards minimizing the root mean square error of the method (using the bound constrained optimization code in MATLAB®, "fminsearchbnd." http://kr.mathworks.com/matlabcentral/fileexchange/8277-fminsearchbnd--fminsearchcon). We randomly divided the available dataset into the datasets for parameter optimization and validation (i.e., validation after optimization). The ratio of the optimization-validation datasets was arbitrarily set to 7:3. Table C1 shows the model parameters and the statistical parameters for the error assessment before and after the parameter optimization. After the optimization, the parameters slightly changed from the default values. However, we still used the default values conservatively since the model results from before and after the optimization were not statistically different in the error assessment.

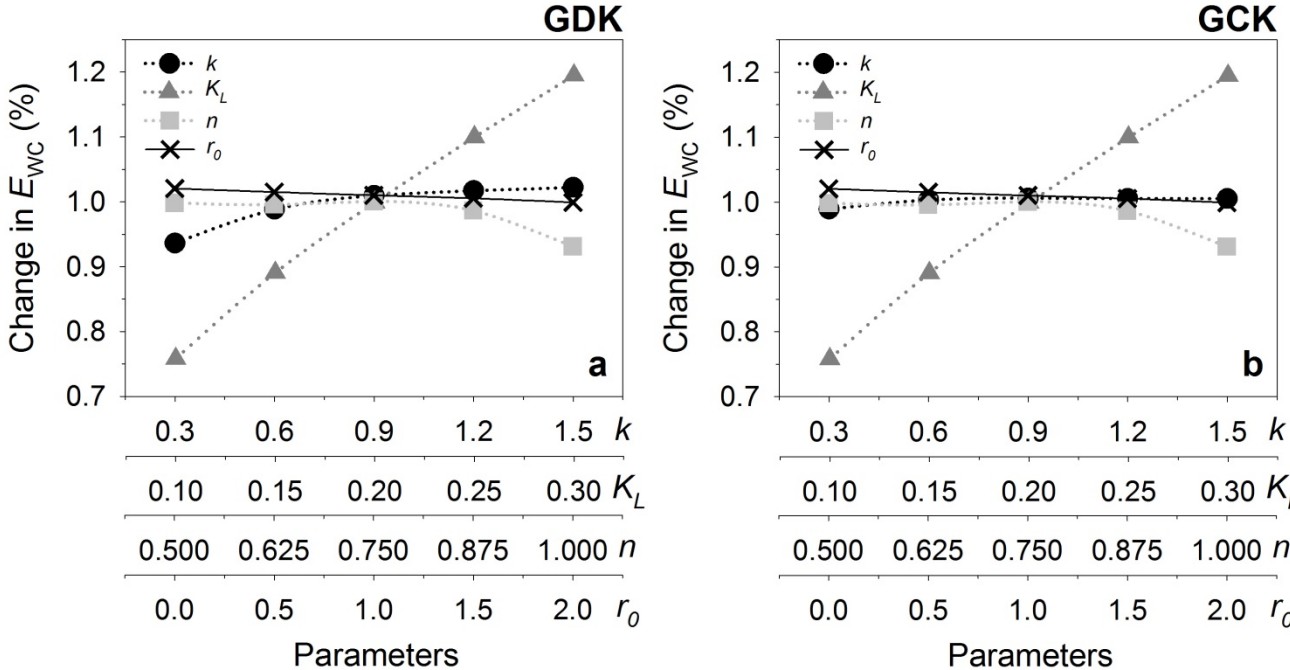

**Figure C1: Sensitivity test of the wet canopy evaporation model to the parameters (i.e., $k$, $K_L$, $n$, and $r_0$).**

**Table C1: Statistical parameters before and after the parameter optimization for the error assessment at the study sites. MBE, MAE, RMSE, and *d* indicate mean bias error, mean absolute error, root mean square error, and index of agreement, respectively. Slope and $r^2$ are from the linear regression analysis. The default values of $K_L$, *n*, and $r_0$ were 0.2 (0.2), 2/3 (2/3), and 2 (2) for the GDK (GCK), respectively. After the optimization, those values changed to 0.1966 (0.2314), 0.7279 (0.6930), and 2 (2) for the GDK (GCK), respectively.**

|  |  |  | MBE | MAE | RMSE | *d* | Slope | $r^2$ |
|---|---|---|---|---|---|---|---|---|
|  |  |  | W m$^{-2}$ | W m$^{-2}$ | W m$^{-2}$ | -- | -- | -- |
| GDK | Optimization dataset ($N = 538$) | before optim. | 10 | 22 | 36 | 0.91 | 0.93 | 0.67 |
| | | after optim. | 10 | 22 | 36 | 0.91 | 0.92 | 0.66 |
| | Validation dataset ($N = 232$) | before optim. | 10 | 19 | 29 | 0.93 | 0.96 | 0.73 |
| | | after optim. | 10 | 19 | 29 | 0.93 | 0.96 | 0.73 |
| GCK | Optimization dataset ($N = 593$) | before optim. | -2 | 24 | 41 | 0.95 | 0.89 | 0.81 |
| | | after optim. | -2 | 24 | 41 | 0.95 | 0.90 | 0.81 |
| | Validation dataset ($N = 256$) | before optim. | -1 | 24 | 38 | 0.95 | 0.87 | 0.81 |
| | | after optim. | -1 | 24 | 38 | 0.95 | 0.88 | 0.81 |

## Appendix D: Modified van Gorsel method

The detailed procedure for selecting the observed nighttime $CO_2$ flux which can represent the actual RE for hilly terrain is proposed by van Gorsel et al. (2009) as follows (the VGF method): (1) calculate the mean diurnal variation of the observed $CO_2$ flux ($F_{CO2\_Obs}$) for a certain period (e.g., 30 days), and identify when the peak of $F_{CO2\_Obs}$ occurs approximately at sunset,

(2) extract $F_{CO2\_Obs}$s one or two hours before and after the time of peak occurrence, (3) exclude the data among the extracted if it rains, or the atmosphere is stable, or the fluctuation of $F_{CO2\_Obs}$ is high, or $F_{CO2\_Obs}$ exceed a certain range, and (4) apply the nighttime correction using the remains. van Gorsel et al. (2009) argued that the RE from the light response curve method ($RE_{LRC}$) can be appropriate criteria for judging abnormal values, and excluded the data, which is smaller than 50% of averaged $RE_{LRC}$ or larger than 200% of that. Notably, they changed the width of the range (e.g., minimum of 50% and a

maximum of 200% to a minimum of 25% and a maximum of 400%) which may be subjective, and confirmed that the results are statistically the same within a 95% confidence interval. However, we discerned that if the result from light response curve method is excessively under/overestimated, the method proposed by van Gorsel et al. (2009) can be affected by the result. Specifically, the result of the nighttime NEE correction is not sensitive to the width of the range, which is used for data filtering criteria, but may be sensitive to the center of the range (i.e., the value of $RE_{LRC}$).

Therefore, we modify the peak values of mean diurnal variation are directly used to estimate the parameters in the RE function (i.e., Lloyd-Taylor equation) based on the basic idea of the VGF method as follows: (1) instead of the mean diurnal variation, use the median diurnal variation which is less affected by the outliers, (2) calculate the median air temperature at the time of peak RE occurrences for the same period, (3) shift the window to calculate the median diurnal variation daily, obtain a sufficient dataset of RE and air temperature (maximum of 30 pairs) for estimating $R_{ref}$ and $E_0$ in the Lloyd-Taylor

equation. This method can be applied independently ($RE_{LRC}$ is not necessary). We call the method the 'modified van Gorsel method ($VGF_m$).' This $VGF_m$ method produces similar results to the original VGF method (with dissimilarity during the monsoon season, which is probably caused by the low data retrieval rate at that time), which supports that the $VGF_m$ method is a good alternative to original VGF method (Fig. D1).

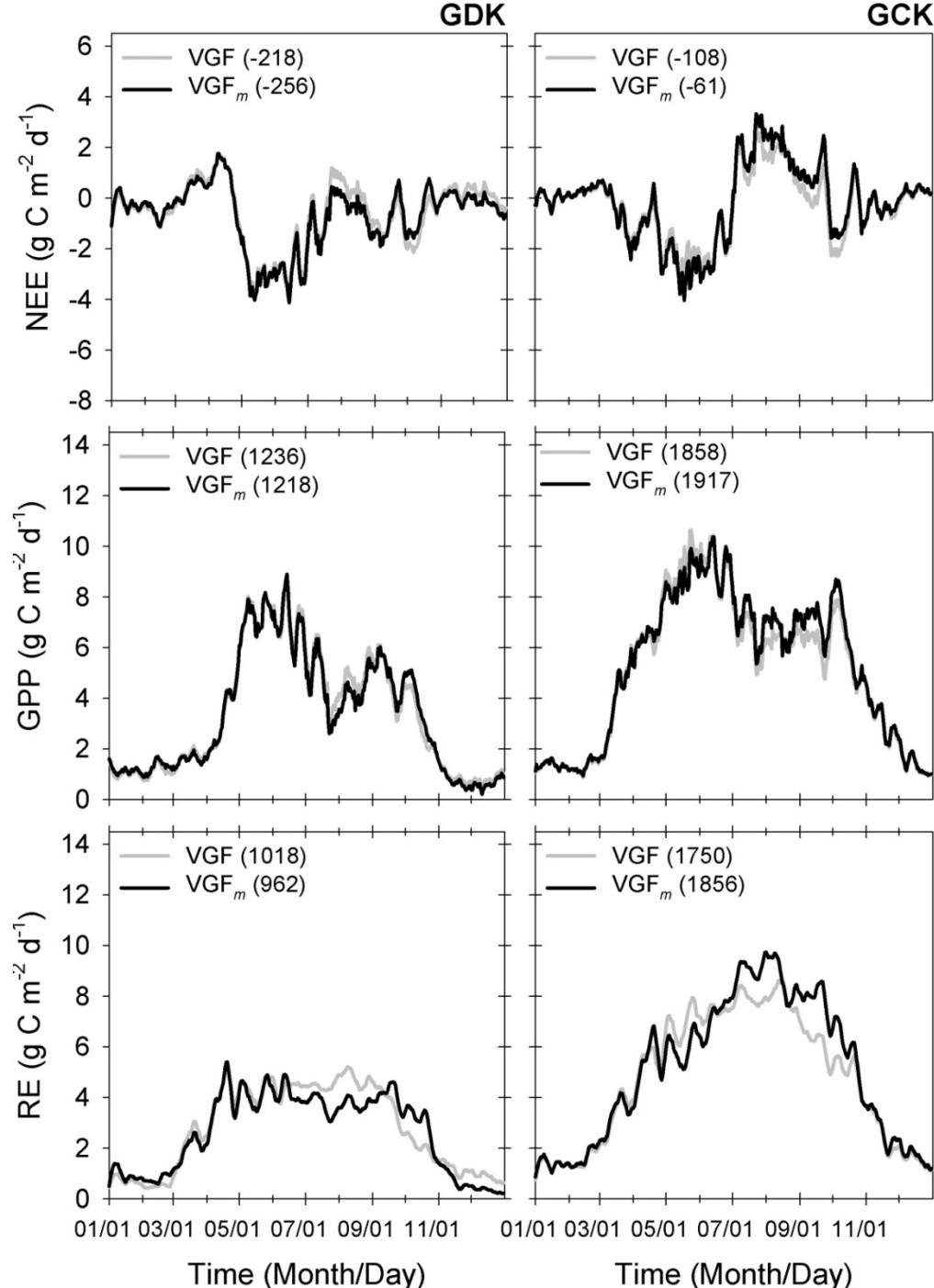

**Figure D1: Seasonal variation of the daily integrated (7-day running mean) net ecosystem exchange (NEE), gross primary production (GPP), and ecosystem respiration (RE) for the GDK and GCK sites in 2008. The VGF and VGF$_m$ mean van Gorsel and modified van Gorsel methods, respectively. The site-specific quality control filter for eliminating the drainage-affected data (Kang et al., 2017) was not applied. The values in parenthesizes mean the annually integrated NEE, GPP, and RE (unit: g C m$^{-2}$ y$^{-1}$).**

## Appendix E: Error assessment

In order to evaluate the latent heat flux under wet canopy condition obtained from the model-stats hybrid method, we compared them against the observed data using four statistical measures, following Willmott and Matsuura (2005). Mean bias error (MBE) is the average of the residuals. Mean absolute error (MAE) is the average of the absolute values of the residuals. A large deviation from zero implies that the estimation generally overestimates or underestimates compared to the observed values. We also considered root mean squared error (RMSE) which is often reported with MAE because RMSE is more sensitive to large errors than MAE.

$$\text{MBE} = \sum \frac{Y_{est} - Y_{obs}}{n} \tag{E1}$$

$$\text{MAE} = \sum \frac{|Y_{est} - Y_{obs}|}{n} \tag{E2}$$

$$\text{RMSE} = \sqrt{\sum \frac{(Y_{est} - Y_{obs})^2}{n}} \tag{E3}$$

MBE, MAE, and RMSE give estimates of the average error, but none of them provides information about the relative size of the average difference. Thus, we further considered an additional index of agreements ($d$), following Willmott (1982):

$$d = 1 - \left[ \frac{\sum (Y_{est} - Y_{obs})^2}{\sum (|Y_{est}'| + |Y_{obs}'|)^2} \right] \tag{E4}$$

where $Y_{est}' = Y_{est} - \overline{Y_{obs}}$ and $Y_{obs}' = Y_{obs} - \overline{Y_{obs}}$ (where overbar is an averaging operator). It ranges from 0 to 1, where 0 is for complete disagreement and 1 for complete agreement between the observation and the estimates. It is both a relative and bounded measure that can be widely applied in order to make cross-comparison between models.