# Peer review of "New gap-filling and partitioning technique for $H_2O$ eddy fluxes measured over forests"

_Biogeosciences, 2017_

## Referee Comment (RC1) · Anonymous Referee #1 · 10 Jul 2017

Dear Authors,

Authors present the new gap-filling and partitioning method of eddy covariance flux over a complex terrain. The methods were combined several previously proposed methods, and were applied to eddy covariance data at Korean forests. I appreciate the work, because FLUXNET community should solve the known problems that authors did. However, in terms of the scope of Biogeosciences journal, the topic is too specific for the eddy covariance technique. I recommend that further modification in terms of the generalization and clarification of the method, especially for the validation and parameterization. Thus, I decide the manuscript as published after the major revision.

[Figure]

Major

Canopy interception model should be validated based on the hydrological measurements or a test data that is from observed data. Without the validation of the model, readers cannot verify the applicability of the model. Incorrect results, due to inappropriate model and/or parameterization, could bias the gap-filled evapotranspiration. Authors need to discuss further model validation. In addition to the validation issue, I cannot follow how authors determined the appropriate model parameters (S, k, n, and g0). If readers want to apply the proposed method, how they will determine the parameters? How is the parameter uncertainties propagate the gap-filled fluxes and partitioned fluxes?

Authors sometime compare the results from the different gap-filling methods or results from previous studies (e.g., Page 11 Lines 24-29). I am not sure which is better, although authors said that the proposed method was better than previous ones without a concrete evidence. Authors must show the additional data for supporting the validity of the method.

The applicability and limitation of this method to other sites, such as tropical forests and grasslands, could be useful for many readers. Currently, only parameter for the two sites were shown as a case study. Further generalization should be required.

Minor

Page 5 Line 27: (Jones, 1993, => (Jones, 1993),

Page 8 Line 26: What is the d statistics?

Fig. 4 : Missing years in x-axis.

Page 31 Line 12 : previous study <= need citation!

Fig. B1: Is this your data? Is the data for your sites or other forests? Further clarification in the caption and citation are required at least.

[Figure]

---

## Referee Comment (RC2) · Anonymous Referee #2 · 15 Jul 2017

The paper proposes mainly a method to fill gaps and partition the water fluxes in evaporation and transpiration in particular in case of wet canopies. In addition it presents a modification in a data filtering method (ustar) that is used for CO2. New methods to partition water fluxes are for sure very interesting and important. For this reason the title is misleading since gapfilling and partitioning of CO2 are not discussed (or only marginally, without real new developments). Improvements are needed in the validation, explanation and uncertainty discussion. I suggest to reorganize it and focus mainly on the more interesting and innovative part (water), adding more sites in order to demonstrate that the approach is valid.

[Figure]

SPECIFIC COMMENTS:

Pages 2-3:The distinction between the ustar filtering and advection methods is not very clear and not fully correct. The ustar filtering has been proposed exactly to filter out data when there is advection so it in not true that it can not be applied when the "drainage flow is at night ". In addition what the authors call "Advection method" is in fact a partitioning method and not a filtering method (e.g. it assumes no advection daytime)

Par 2.3.2: there are a lot of parameters in the proposed model but it is not clear how they are estimated and also which is the associated uncertainty. Also it should be verified if the parameters are valid in different conditions (to test the "everywhere, all of the time" proposed by the authors P3L18). At the moment this is far to be demonstrated.

I find the section 2.3.3 not clear in the second part where the application of the model is reported. More information and a clear description of the procedure are needed.

Section 2.4.1 reports the NEE processing but the level of details is not sufficient to understand what is done (how are the parameter estimated? How is the ustar threshold calculated?). In addition three methods are presented as three independent approached but it is not explained if, when the light response curve method is applied, the data below the ustar threshold are removed (in this case they are not independent, if not removed that it is an error because it is known that the data are not correct). That said, I also find this part not relevant for the paper that is more focussed on H2O.

P7L28-30: authors should explain why the MPT method is "inappropriate for hilly terrain sites" while the ustar filtering (and so also the MPT) has been developed specifically to filter out the advection.

P7L31: the fact that near sunset the drainage has not yet completely developed must be proved.

P8L1-3: how can the authors be sure that the average nighttime flux in the two windows
is not different for biological reasons? Or because of the uncertainty in the storage term? Or because a different wind direction distribution (i.e. footprint) is present in the two time windows?

P8L4-5: it is not clear what is proposed. If the two windows are different then all the data in the second window are removed? And ustar filtering is applied to what?

P8L18-21: is the model validation made using a leave-one-out method or using independent dataset?

Section 3.1.2: the comparison of the methods is no a validation and no strong conclusions can be derived from this. In addition, in the Figure 4 uncertainty is not considered to understand how much (if) the two approaches are significantly different. The only way to prove this is to add other sites, for example where close path IRGAs with heated tube are used (so that ET measurements are ok also with rain) and then apply the methods to artificial gaps.

Section 3.2.1: in Figure 5 the comparison of measurements before and after sunset is used to show the presence of drainage. However it is not clear if the data shown in the plot are filtered by ustar (they should otherwise the plot is biased by a known issue). The Table to add is the 2 and not the 1. Figure 6 doesn't show the original method and how much it is different respect to the modified. In addition it is not a validation because however compared with other models/methods. Figure 7: how was heterotrophic respiration measured?

Section 4: I find this section, although with some nice and interesting aspects, out of scope respect to the rest of the paper.

Appendix A, lines 22-23: the main reason of the minor effect of the storage method in ET is that the fluxes are always very low at night, when the storage component is important.

---

## Author Comment (AC1) · 7 Sep 2017

We very much appreciate the reviewer's critical yet constructive comments, allowing us to reassess and improve our manuscript. Please see the below for the authors' reply

Authors present the new gap-filling and partitioning method of eddy covariance flux over a complex terrain. The methods were combined several previously proposed methods, and were applied to eddy covariance data at Korean forests. I appreciate the work, because FLUXNET community should solve the known problems that authors did. However, in terms of the scope of Biogeosciences journal, the topic is too specific for the eddy covariance technique. I recommend that further modification in terms of the generalization and clarification of the method, especially for the validation and parameterization. Thus, I decide the manuscript as published after the major revision.

Major

Canopy interception model should be validated based on the hydrological measurements or a test data that is from observed data. Without the validation of the model, readers cannot verify the applicability of the model. Incorrect results, due to inappropriate model and/or parameterization, could bias the gap-filled evapotranspiration. Authors need to discuss further model validation. In addition to the validation issue, I cannot follow how authors determined the appropriate model parameters (S, k, n, and g0). If readers want to apply the proposed method, how they will determine the parameters? How is the parameter uncertainties propagate the gap-filled fluxes and partitioned fluxes?

Response: We fully agree with the reviewer's comments. We will revise the manuscript as follows.

(1) Add more detailed explanations how we can obtain the parameters from our field measurement (mainly from the flux tower) and introduce alternative ways (e.g., using MODIS product)

(2) Add a section for sensitivity analysis of the proposed method similarly to that from Shi et al. (2010), and identify the parameters which significantly affect the gap-filling and partitioning results.

[Figure]

Fig. 1. Sensitivity analysis of the revised analytical model: influence of parameters $S$ (canopy storage capacity), $c$ (canopy cover), $p_t$ (proportion of rain diverted to stem flow) and $S_t$ (trunk storage capacity), and of climate variables $E$ (mean evaporation rate during rainfall) and $R$ (mean rainfall rate) (copied from Shi et al., 2010).

(3) Add a section for the (sensitive) parameters optimization. We should maximize the validity of (a small number of) the observed $H_2O$ flux data under wet canopy condition. In the original manuscript, we used all available data under wet canopy condition to validate the method. In the revised manuscript, we will divide the available dataset into the datasets for parameter optimization and validation (i.e., validation after optimization). The ratio of the optimization-validation datasets may be 7:3. Such strategy can improve the applicability of the method (i.e., generalization).

Shi, Z., Wang, Y., Xu, L., Xiong, W., Yu, P., Gao, J. & Zhang, L. (2010) Fraction of incident rainfall within the canopy of a pure stand of Pinus armandii with revised Gash model in the Liupan Mountains of China. *Journal of Hydrology, 385,* 44-50.

Authors sometime compare the results from the different gap-filling methods or results from previous studies (e.g., Page 11 Lines 24-29). I am not sure which is better, although authors said that the proposed method was better than previous ones without a concrete evidence.

Response: We agree with the reviewer's comment. The underestimation of the gap-filled $H_2O$ flux

under wet canopy condition from the conventional marginal distribution sampling (MDS) method has been shown by the comparison with the sum of energy flux components except latent heat flux (= net radiation + sensible heat flux + storage flux) in our previous study (Kang et al. 2012, the results from the proposed model-stats hybrid method (MSH) displayed the mirrored patterns of the sum of the other energy budget components, while the results from the MDS were ~ 0, see the below figure).

[Figure]

Fig. 2. Diurnal variation of net radation ($R_N$), latent heat flux ($\lambda E$), sensible heat flux ($SH$), the sum of three energy components ( = $R_N$+ $S_E$ + $SH$; where $S_E$ is energy storage), and wet canopy evaporation simulated by the modified lookup table method ($\lambda E_{WC\_MLT}$) and the algorithm of VIC LSM ($\lambda E_{WC\_VIC}$) at the GDK and the GCK sites. The shaded area represents the period of wet canopy condition. (copied Kang et al., 2012)

Based on the previous finding (i.e., Kang et al., 2012) and the validation results (section 3.1.1 in the manuscript), we argued that the proposed method was better.

The best evidence which supports the proposed method was better than previous one (i.e., in some year, rainfall increased evapotranspiration (it means that the increased wet canopy evaporation exceeded the decreased transpiration due to rainfall), and the underestimation of ET from the previous method especially in the summer of 2007 due to the unaccounted wet canopy

evaporation) may be another actual (flux) measurement. If another actual measurement can be obtained easily, such gap-filling and partitioning would not be a scientific issue. Fortunately, there was the previous study which reported the runoff from the forest catchment (Choi et al. 2011).

We will revise the manuscript as follows.

(1) Add the paragraph which explains that the conventional gap-filling method underestimates $H_2O$ flux under wet canopy condition (i.e., a more detailed summary of our previous study, Kang et al., 2012).

(2) Add the sentences which can support the proposed method was better than previous one (e.g., Page 11 Lines 24-29): (1) the ratio of the runoff and the precipitation (adapted from Choi et al. 2011) in 2007 was the lowest (0.60 in 2007, 0.69±0.06 in the other years, i.e., the ratio of the ET to the precipitation can be the highest), while the global radiation (main controlling factor of transpiration) was the lowest (4.52 GJ $m^{-2}$ in 2007, 4.77±0.08 GJ $m^{-2}$ in the other years) due to the longest rainfall duration, (2) it was identified that the interannual variabilities of the estimated catchment scale annual ET (i.e., precipitation – runoff) and ET from the MDS method occurred in opposite directions (similarly to transpiration from the MHS method).

Kang, M., Kwon, H., Cheon, J. H., & Kim, J. (2012). On estimating wet canopy evaporation from deciduous and coniferous forests in the Asian monsoon climate. *Journal of Hydrometeorology*, *13*(3), 950-965.

Choi, H. T. (2011). Effect of Forest Growth and Thinning on the Long-term Water Balance in a Coniferous Forest. *Korean Journal of Agricultural and Forest Meteorology*, *13*(4), 157-164.

Authors must show the additional data for supporting the validity of the method. The applicability and limitation of this method to other sites, such as tropical forests and grasslands, could be useful for many readers. Currently, only parameter for the two sites were shown as a case study. Further generalization should be required.

Response: We think the generalization of the method can be augmented by providing the parameter optimization procedure using available flux data under wet canopy condition. We also

argue that this is better than the validation using other datasets because the parameters may be site specific (i.e., more validation does not fully guarantee the proposed method works properly everywhere). The proposed method can be applied to tropical forests because tropical forests also share three properties of temperate forests (i.e., extensive, dense, and tall). However, applying the methods to grasslands may need further validation. We will mention these in the manuscript.

Minor

Page 5 Line 27: (Jones, 1993, => (Jones, 1993),

Response: We will correct as suggested.

Page 8 Line 26: What is the d statistics?

Response: The index of agreements ($d$, Willmott, 1982) is defined as follows:

$$d = 1 - \left[ \frac{\sum_{i=1}^{N} (Y_{\text{est}_i} - Y_{\text{obs}_i})^2}{\sum_{i=1}^{N} (|Y_{\text{est}_i}'| + |Y_{\text{obs}_i}'|)^2} \right]$$

where $Y_{\text{est}_i}' = Y_{\text{est}_i} - \overline{Y_{\text{obs}}}$ and $Y_{\text{obs}_i}' = Y_{\text{obs}_i} - \overline{Y_{\text{obs}}}$ (where the overbar is an averaging operator). The index ranges from 0 to 1, where 0 represents a complete disagreement, and 1 represents the complete agreement between the observation and the estimates. This index is both a relative and bounded measure that can be widely applied to make cross-comparisons between models.

We will add a section for the error assessment. In the section, we will define each error assessment term mathematically as above.

Fig. 4 : Missing years in x-axis.

Response: We will correct as suggested.

Page 31 Line 12 : previous study <= need citation!

Response: We already provided the citations in Table B1.

Fig. B1: Is this your data? Is the data for your sites or other forests? Further clarification in the caption and citation are required at least.

Response: We already provided the citations in Table B1. We will modify the caption to avoid such confusion.

---

## Author Comment (AC2) · 7 Sep 2017

We very much appreciate the reviewer's critical yet constructive comments, allowing us to sharpen our focus and improve our manuscript. Please see the below for our reply.

The paper proposes mainly a method to fill gaps and partition the water fluxes in evaporation and transpiration in particular in case of wet canopies. In addition it presents a modification in a data filtering method (ustar) that is used for CO2. New methods to partition water fluxes are for sure very interesting and important. For this reason the title is misleading since gapfilling and partitioning of CO2 are not discussed (or only marginally, without real new developments). Improvements are needed in the validation, explanation and uncertainty discussion. I suggest to reorganize it and focus mainly on the more interesting and innovative part (water), adding more sites in order to demonstrate that the approach is valid.

Response: We partially agree with the reviewer's comments. More validation, explanation, and uncertainty of the proposed method (especially water flux gap-filling and partitioning part) are required to improve the manuscript. However, we do not agree to remove the $CO_2$ flux part. Because we argue that (1) nighttime flux data filtering is a part of gap-filling and partitioning of $CO_2$ flux, (2) there is still a lack of effort to link carbon and water fluxes even though they are strongly connected by stomata, and we typically measure the both simultaneously. The gap-filling (and partitioning) of flux data is a kind of interpolation (and extrapolation) of available observed fluxes using the relationship between the fluxes and its drivers. For identifying the relationship correctly, selecting data (i.e., filtering data which cannot represent the phenomena/signals we want/expect) is the most important first step. We will add a paragraph for explaining (briefly) the nature of gap-filling and partitioning of eddy covariance (EC) flux data in the introduction section. Also, we argue that the last message of this paper is also important: There is a common characteristic between the proposed methods, i.e., two existing methods are merged into a new method. Such a strategy strengthens the strength and makes up for the weakness of the original methods. It also results in better applicability. It will contribute to the standardization of eddy covariance data processing.

SPECIFIC COMMENTS:

Pages 2-3:The distinction between the ustar filtering and advection methods is not very clear and not fully correct. The ustar filtering has been proposed exactly to filter out data when there is

advection so it in not true that it can not be applied when the "drainage flow is at night ". In addition what the authors call "Advection method" is in fact a partitioning method and not a filtering method (e.g. it assumes no advection daytime)

Response: We agree with the reviewer's comments. We think our original explanations make readers confused as reviewer's comments. To avoid such confusion, we will add a paragraph for explaining (briefly) the general procedure of $CO_2$ EC flux partitioning (i.e., selecting data to identify the relationship between nighttime $CO_2$ flux and its driver, identifying the relationship, extrapolating the relationship to daytime (or nighttime)) in the introduction section. We will also define the related terminologies in the paragraph.

Par 2.3.2: there are a lot of parameters in the proposed model but it is not clear how they are estimated and also which is the associated uncertainty. Also it should be verified if the parameters are valid in different conditions (to test the "everywhere, all of the time" proposed by the authors P3L18). At the moment this is far to be demonstrated.

Response: We fully agree with the reviewer's comments. We will revise the manuscript as follows.

(1) Add more detailed explanations how we can obtain the parameters from our field measurement (mainly from the flux tower) and introduce alternative ways (e.g., using MODIS product)

(2) Add a section for sensitivity analysis of the proposed method similarly to that from Shi et al. (2010), and identify the parameters which significantly affect the gap-filling and partitioning results.

[Figure]

Fig. 1. Sensitivity analysis of the revised analytical model: influence of parameters $S$ (canopy storage capacity), $c$ (canopy cover), $p_t$ (proportion of rain diverted to stem flow) and $S_t$ (trunk storage capacity), and of climate variables $E$ (mean evaporation rate during rainfall) and $R$ (mean rainfall rate) (copied from Shi et al., 2010).

(3) Add a section for the (sensitive) parameters optimization. We should maximize the validity of (a small number of) the observed $H_2O$ flux data under wet canopy condition. In the original manuscript, we used all available data under wet canopy condition to validate the method. In the revised manuscript, we will divide the available dataset into the datasets for parameter optimization and validation (i.e., validation after optimization). The ratio of the optimization-validation datasets may be 7:3. Such strategy can improve the applicability of the method (i.e., generalization for '*everywhere, all of the time*').

Shi, Z., Wang, Y., Xu, L., Xiong, W., Yu, P., Gao, J. & Zhang, L. (2010) Fraction of incident rainfall within the canopy of a pure stand of Pinus armandii with revised Gash model in the Liupan Mountains of China. *Journal of Hydrology*, *385*, 44-50.

I find the section 2.3.3 not clear in the second part where the application of the model is reported. More information and a clear description of the procedure are needed.

Response: We agree with the reviewer's comments. In the original manuscript, the detailed explanations of the model were omitted to avoid self-plagiarism (authors' previous study, Kang et

al., 2012). We will revise the section 2.3.3 (add the detailed information).

Kang, M., Kwon, H., Cheon, J. H., & Kim, J. (2012). On estimating wet canopy evaporation from deciduous and coniferous forests in the Asian monsoon climate. *Journal of Hydrometeorology*, *13*(3), 950-965.

Section 2.4.1 reports the NEE processing but the level of details is not sufficient to understand what is done (how are the parameter estimated? How is the ustar threshold calculated?). In addition three methods are presented as three independent approached but it is not explained if, when the light response curve method is applied, the data below the ustar threshold are removed (in this case they are not independent, if not removed that it is an error because it is known that the data are not correct). That said, I also find this part not relevant for the paper that is more focussed on H2O.

Response: We mostly agree with the reviewer's comments (except the last argument). We will rewrite the section 2.4.1 including the following contents.

We adapted 0.3 m $s^{-1}$ of $u*$ threshold from Kang et al. (2014 and 2017). We had checked the dependency of nighttime $CO_2$ flux on friction velocity during the growing and dormant seasons (Fig. 3). 0.3 m $s^{-1}$ of $u*$ threshold can be determined during the growing season, while it is hard to clearly decide a threshold during the dormant season for the both sites. But, the threshold during the dormant season would be smaller than 0.3 m $s^{-1}$. Therefore, we applied the constant threshold of 0.3 m $s^{-1}$ for the sites.

[Figure]

[Figure]

Fig. 3. The dependency of respiration of ecosystem (i.e., nighttime net ecosystem exchange) on friction velocity for the uphill (GDK) and downhill (GCK) sites during the growing (a, DOY 121-300 for the uphill and DOY 91-300 for the downhill) and dormant (b, non-growing season) seasons. (adapted from Kang et al., 2014 and 2017).

Each method is independent each other. In cases of the light response curve method and the advection-based method, the nighttime $CO_2$ fluxes were filtered out when the observed nighttime $CO_2$ fluxes were underestimated out of the 95% confidence interval of the ecosystem respiration model (i.e., Lloyd and Taylor equation).

P7L28-30: authors should explain why the MPT method is "inappropriate for hilly terrain sites" while the ustar filtering (and so also the MPT) has been developed specifically to filter out the advection.

Response: We annexed a proviso: inappropriate for hilly terrain sites "that are usually affected by drainage flow." The advection-based method was developed for such sites (van Gorsel et al., 2007, 2008, and 2009). Figure 4 (copied from van Gorsel et al., 2007) (and Figure 5 which is adapted from Kang et al., 2017, our previous study for the same study sites) shows why it is hard to apply the $u^*$ filtering method for such sites. During the nighttime except near sunset, the $CO_2$ fluxes were close to 0 and much smaller compared with the data from the other independent observation (i.e., chamber), due to drainage flows. It suggests that we should consider not only $u^*$ but also time (when the drainage flow is fully developed) for nighttime $CO_2$ flux filtering. We will add such explanation in the introduction and method sections.

[Figure]

Fig. 4. Mean daily course [2001–2005] of the turbulent flux of $CO_2$ (black line), the sum of eddy flux and change in storage term (gray line) and the soil temperature at 0.02 m (dash dotted line). The dotted line represents total nighttime respiration derived from chamber measurements (copied from van Gorsel et al., 2007).

[Figure]

Fig. 5. Mean diurnal variations of the observed $CO_2$ fluxes and storage terms for the uphill (GDK)

and downhill (GCK) sites during the growing season (i.e., June–September). The error bars indicate the standard deviation for each half hour. The shaded areas represent nighttime, and the unshaded areas represent daytime.

van Gorsel, E., Leuning, R., Cleugh, H. A., Keith, H., & Suni, T. (2007). Nocturnal carbon efflux: Reconciliation of eddy covariance and chamber measurements using an alternative to the u*-threshold filtering technique. *Tellus B*, *59*(3), 397-403.

van Gorsel, E., Leuning, R., Cleugh, H. A., Keith, H., Kirschbaum, M. U., & Suni, T. (2008). Application of an alternative method to derive reliable estimates of nighttime respiration from eddy covariance measurements in moderately complex topography. *Agricultural and Forest Meteorology*, *148*(6), 1174-1180.

van Gorsel, E., Delpierre, N., Leuning, R., Black, A., Munger, J. W., Wofsy, S., ... & Chen, B. (2009). Estimating nocturnal ecosystem respiration from the vertical turbulent flux and change in storage of $CO_2$. *Agricultural and Forest Meteorology*, *149*(11), 1919-1930.

Kang, M., Ruddell, B. L., Cho, C., Chun, J., & Kim, J. (2017). Identifying $CO_2$ advection on a hill slope using information flow. *Agricultural and Forest Meteorology*, *232*, 265-278.

P7L31: the fact that near sunset the drainage has not yet completely developed must be proved.

Response: It had been proven using bulk Richardson number, $CO_2$ concentration profile, $CO_2$ flux measured by chamber method (van Gorsel et al., 2007, 2008) and information flow between the uphill and downhill in the previous studies (Kang et al., 2017). We will add the details in the manuscript.

van Gorsel, E., Leuning, R., Cleugh, H. A., Keith, H., & Suni, T. (2007). Nocturnal carbon efflux: Reconciliation of eddy covariance and chamber measurements using an alternative to the u*-threshold filtering technique. *Tellus B*, *59*(3), 397-403.

van Gorsel, E., Leuning, R., Cleugh, H. A., Keith, H., Kirschbaum, M. U., & Suni, T. (2008). Application of an alternative method to derive reliable estimates of nighttime respiration from eddy covariance measurements in moderately complex topography. *Agricultural and Forest*

*Meteorology, 148*(6), 1174-1180.

Kang, M., Ruddell, B. L., Cho, C., Chun, J., & Kim, J. (2017). Identifying $CO_2$ advection on a hill slope using information flow. *Agricultural and Forest Meteorology, 232*, 265-278.

P8L1-3: how can the authors be sure that the average nighttime flux in the two windows is not different for biological reasons? Or because of the uncertainty in the storage term? Or because a different wind direction distribution (i.e. footprint) is present in the two time windows?

Response: Wind directions (mountain wind) were same. And, we did not find any driver which makes a difference between the averages nighttime flux in the two time windows except ecosystem temperature. So, we compared the averages after normalizing the flux measurements using the temperature response function (i.e., Lloyd and Taylor equation) as same as the original MPT method proposed by Gu et al. (2005). We will add this argument in the manuscript.

Gu, L., Falge, E. M., Boden, T., Baldocchi, D. D., Black, T. A., Saleska, S. R., ... & Xu, L. (2005). Objective threshold determination for nighttime eddy flux filtering. *Agricultural and Forest Meteorology, 128*(3), 179-197.

P8L4-5: it is not clear what is proposed. If the two windows are different then all the data in the second window are removed? And ustar filtering is applied to what?

Response: Yes, all the data in the second time window are removed. The $u^*$ filtering is applied to the first time window. We will revise the sentence.

P8L18-21: is the model validation made using a leave-one-out method or using independent dataset?

Response: Originally, the model was fully independent. Following the reviewer's comment and reminding the nature of gap-filling and partitioning of EC flux data, we should maximize the validity of (a small number of) the observed $H_2O$ flux data under wet canopy condition. In the original manuscript, we used all available data under wet canopy condition to validate the method.

In the revised manuscript, we will divide the available dataset into the datasets for parameter optimization and validation (i.e., validation after optimization). The ratio of the optimization-validation datasets may be 7:3.

Section 3.1.2: the comparison of the methods is no a validation and no strong conclusions can be derived from this. In addition, in the Figure 4 uncertainty is not considered to understand how much (if) the two approaches are significantly different. The only way to prove this is to add other sites, for example where close path IRGAs with heated tube are used (so that ET measurements are ok also with rain) and then apply the methods to artificial gaps.

Response: The difference between the two methods and new information provided by the newly proposed method motivate readers to apply the new method. If another actual measurement (like the reviewer mentioned, i.e., close path IRGAs with heated tube are used) can be obtained easily, such gap-filling and partitioning would not be a scientific issue. Even though the number of observed data under wet canopy condition is small (in this study), we can validate and optimize the model. For quantifying the uncertainty of the model, we apply Monte Carlo simulations for sampling the optimization and validation dataset similarly to Richardson and Hollinger, 2007. We will add this discussion in the manuscript.

Richardson, A. D., & Hollinger, D. Y. (2007). A method to estimate the additional uncertainty in gap-filled NEE resulting from long gaps in the $CO_2$ flux record. *Agricultural and Forest Meteorology*, *147*(3), 199-208.

Section 3.2.1: in Figure 5 the comparison of measurements before and after sunset is used to show the presence of drainage. However it is not clear if the data shown in the plot are filtered by ustar (they should otherwise the plot is biased by a known issue).

Response: The figure shows the results after applying the $u^*$ filtering. It also showed that determining the $u^*$ threshold is difficult when drainage is fully developed due to ~0 of $CO_2$ fluxes. We will consider adding the data before applying the $u^*$ filtering in the figure.

The Table to add is the 2 and not the 1.

Response: We will correct as suggested.

Figure 6 doesn't show the original method and how much it is different respect to the modified. In addition it is not a validation because however compared with other models/methods.

Response: We will consider adding the results from the original MPT method in the figure. The other method was developed in our previous research, Kang et al. (2017). Currently (without other independent measurements such as chamber method), it was the best results which have been published in the peer-reviewed journal.

Kang, M., Ruddell, B. L., Cho, C., Chun, J., & Kim, J. (2017). Identifying $CO_2$ advection on a hill slope using information flow. *Agricultural and Forest Meteorology, 232*, 265-278.

Figure 7: how was heterotrophic respiration measured?

Response: Heterotrophic respiration was not considered. We will mention the limitation of this comparison, or estimate (and consider) the heterotrophic respiration based on the literature review.

Section 4: I find this section, although with some nice and interesting aspects, out of scope respect to the rest of the paper.

Response: This section showed readers why we should do ET partitioning.

Appendix A, lines 22-23: the main reason of the minor effect of the storage method in ET is that the fluxes are always very low at night, when the storage component is important.

Response: We will revise as suggested.

---

## Author Response (AR2)

We very much appreciate the reviewer's critical yet constructive comments, allowing us to reassess and improve our manuscript. Please see the below for the authors' reply.

The authors did a good job trying to improve the quality and answer the reviewers questions. That said I still see some major issue that prevent the publication of this article:

1) I still strongly think that the CO2 part of the paper is minor and shadowing the interesting analysis on water. The authors refer to CO2 partitioning that is not discussed in the paper (just an equation, already published, is applied and no gapfilling) and also the very important link between C and H2O is however not addressed in the paper. So I still think that the CO2 part is just marginal (and with issues, see below)

2) In the Section 2.4.1 it is still not clear if the LRC is parameterized using all the data or only data with ustar above the threshold. In the first case this is wrong, because these data are known to be underestimated, in the second case the method would be not any more independent by FVF. However as stated above I suggest to remove this section and may be write a separate paper on CO2 where all these aspects are analysed. In fact the overall discussion is in my opinion not robust enough, ignoring uncertainty (are they really different? Did you check the overall uncertainty?) and proposing a method that removes a lot of data (see P9, L16-17) so increasing the uncertainty in particular as function of the gapfilling method used. Also it looks like there is still confusion on the ustar method and advection: working or not working ustar filtering was proposed and developed for advection and hilly terrains, saying that is not appropriate is wrong. May be not sufficient in some cases but again this needs to be shown and discussed.

Response (comments 1 and 2): The $CO_2$ flux part of the manuscript was deleted as suggested.

3) I still think that the method should be part of a real validation and not a comparison between models or the use of measurements that are considered wrong in the paper (open path during rain). It means (referring to comments on Section 3.1.2 and answers) that the authors should find dataset were a close or closed path IRGA with heated tube is used and then apply the method developed to these data (e.g. adding artificial gaps) to validate the model and estimate the uncertainty. There are 600 sites in FLUXNET, a dataset like this is for sure available.

Response: Recently, we conducted a closed-path IRGA for the one of the study sites (i.e., GDK) with the original open-path EC system. We conducted the further evaluation using the data from the closed-path EC system. As we expected, the data retrieval rate under wet canopy condition of the closed-path EC system was 50% higher than that of the open-path EC system. However, there were still missing data for the considerable period despite the closed-path gas analyzer was conducted. The missing was mainly caused by the malfunction of the sonic anemometer and the unsatisfactory conditions for EC measurement (e.g., nonstationary and unfavorable turbulent developed conditions). Unlikely to photosynthesis and transpiration, wet canopy evaporation is strongly connected to the wet canopy evaporation which occurred in the previous time step (i.e., the wet canopy evaporation in the previous time step determines the intercepted canopy water. The model is a numerical model, not an analytical model). Because of such reasons, it is challengeable to validate (and estimate the uncertainty) the method as suggested (i.e., adding artificial gaps and gap-filling). Instead of that, we conducted the comparison between the model result and the observation from the closed-path EC system. The results of the error assessment were similar to those from the comparison between the model result and the observation from the open-path EC system. We added those results in the Appendix C.

The wet canopy evaporation occurs during and following rain events. The both, the open-path EC system and closed-path system, can measure some (NOT all) of the wet canopy evaporation. The latter has not only the advantage but also the disadvantage i.e., more robust under wet canopy condition (resulting higher data retrieval rate) and the underestimation due to tube attenuation in high RH condition. Additionally, there is no sonic anemometer which can measure perfectly and continuously during rain events (it also fails when the transducer probe is contaminated by rain drops).

Thus, we argue that (1) the dataset from a closed-path EC system is also not a perfect for assessing wet canopy evaporation; (2) the difference of data quality between the open-path and the closed-path is minor and only difference is the periods that instruments work properly; (3) an appropriate wet canopy evaporation gap-filling method is also necessitated in the case of the measurement using a closed path EC system, like the case of an open-path EC system. Related to the reviewer's suggestion, we also will do our efforts to measure evapotranspiration during and following rain events reliably and continuously and hope to estimate the uncertainty of our evapotranspiration gap-filling and partitioning methodology following the reviewer's suggestion as further study.

[revised manuscript text omitted]

$$\text{MAE} = \sum \frac{\left| Y_{\text{est}} - Y_{\text{obs}} \right|}{n} \tag{DE2}$$

$$\text{RMSE} = \sqrt{\sum \frac{(Y_{\text{est}} - Y_{\text{obs}})^2}{n}} \tag{DE3}$$

MBE, MAE, and RMSE give estimates of the average error, but none of them provides information about the relative size of the average difference. Thus, we further considered an additional index of agreements (*d*), following Willmott (1982):

$$d = 1 - \left[ \frac{\sum (Y_{\text{est}} - Y_{\text{obs}})^2}{\sum (\left| Y_{\text{est}}' \right| + \left| Y_{\text{obs}}' \right|)^2} \right] \tag{DE4}$$

where $Y_{\text{est}}' = Y_{\text{est}} - \overline{Y_{\text{obs}}}$ and $Y_{\text{obs}}' = Y_{\text{obs}} - \overline{Y_{\text{obs}}}$ (where overbar is an averaging operator). It ranges from 0 to 1, where 0 is for complete disagreement and 1 for complete agreement between the observation and the estimates. It is both a relative and bounded measure that can be widely applied in order to make cross-comparison between models.

---

## Author Response (AR3)

We very much appreciate the editor's critical yet constructive comments, allowing us to reassess and improve our manuscript. Please see the below for the authors' reply.

I accept the current version of the manuscript pending to the following technical corrections (not requiring further editor assessment):

(1) p. 3, l. 22, p. 3, l. 24, p. 4, l. 1: (and CO2)

Response: corrected as suggested

The following suggestions are meant to make the structure of the paper conform with the classical structure of scientific publications:

(2) p. 7: I suggest section 3 should be entitled "Results and discussion"

Response: corrected as suggested

(3) p. 9: following on with (2) I suggest section 4.1 to be renumbered as 3.3 and entitled "Application: Wavelet coherence ...."

Response: corrected as suggested

(4) p. 9: to fit with the above change, I suggest to remove the first line on p. 9 and start the sentence in line 2 with "In the following we illustrate ...."

Response: corrected as suggested

(5) p. 10: section 4.2 should be numbered 3.4 and the title changed to "Application: "Water use ..."

Response: corrected as suggested

(6) p. 10: section 5 should be numbered as section 4

Response: corrected as suggested

[revised manuscript text omitted]